# Regression with reject option and application to $k$NN

**Ahmed Zaoui**[*]
LAMA, Université Gustave Eiffel
ahmed.zaoui@univ-eiffel.fr

**Christophe Denis**
LAMA, Université Gustave Eiffel
MIA-Paris, AgroParisTech, INRAE, Université Paris-Saclay
christophe.denis@univ-eiffel.fr

**Mohamed Hebiri**
LAMA, Université Gustave Eiffel
CREST, ENSAE, Institut Polytechnique de Paris
mohamed.hebiri@univ-eiffel.fr

## Abstract

We investigate the problem of regression where one is allowed to abstain from predicting. We refer to this framework as *regression with reject option* as an extension of classification with reject option. In this context, we focus on the case where the rejection rate is fixed and derive the optimal rule which relies on thresholding the conditional variance function. We provide a semi-supervised estimation procedure of the optimal rule involving two datasets: a first *labeled* dataset is used to estimate both regression function and conditional variance function while a second *unlabeled* dataset is exploited to calibrate the desired rejection rate. The resulting predictor with reject option is shown to be almost as good as the optimal predictor with reject option both in terms of risk and rejection rate. We additionally apply our methodology with $k$NN algorithm and establish rates of convergence for the resulting $k$NN predictor under mild conditions. Finally, a numerical study is performed to illustrate the benefit of using the proposed procedure.
**Keywords:** Regression; Regression with reject option; $k$NN; Predictor with reject option.

## 1 Introduction

Confident prediction is a fundamental problem in statistical learning for which numerous efficient algorithms have been designed, *e.g.,* neural-networks, kernel methods, or $k$-Nearest-Neighbors ($k$NN) to name a few. However, even state-of-art methods may fail in some situations, leading to bad decision-making. Obvious damageable incidences of an erroneous decision may occur in several fields such as medical diagnosis, where a wrong estimation can be fatal. In this work, we provide a novel statistical procedure designed to handle these cases. In the specific context of regression, we build a prediction algorithm that allows to abstain from predicting when the doubt is too important. As a generalization of the classification with reject option setting [4, 5, 6, 11, 13, 16, 22], this framework is naturally referred to as *regression with reject option*. In the spirit of [6], we opt here for a strategy where the predictor can abstain up to a fraction $\varepsilon \in (0,1)$ of the data. The merit of this approach is that it allows human action on the proportion of the data where the prediction is too difficult while

---

[*]ahmed.zaoui@univ-eiffel.fr

standard machine learning algorithms can be exploited to perform the predictions on the other fraction of the data. The difficulty to address a prediction is then automatically evaluated by the procedure. From this perspective, this strategy may improve the efficiency of the human intervention.

In this paper, we investigate the regression problem with reject option when the rejection (or abstention) rate is controlled. Specifically, we provide a statistically principled and computationally efficient algorithm tailored to this problem. We first formally define the regression with reject option framework, and explicitly exhibit the optimal predictor with bounded rejection rate in Section 2. This optimal rule relies on a thresholding of the conditional variance function. This result is the bedrock of our work and suggests the use of a plug-in approach. We propose in Section 3 a two-step procedure which first estimates both the regression function and the conditional variance function on a first *labeled* dataset and then calibrates the threshold responsible for abstention using a second *unlabeled* dataset. Under mild assumptions, we show that our procedure performs as well as the optimal predictor both in terms of risk and rejection rate. We emphasize that our procedure can be exploited with any off-the-shell estimator. As an example we apply in Section 4 our methodology with the $k$NN algorithm for which we derive rates of convergence. Finally, we perform numerical experiments in Section 5 which illustrate the benefits of our approach. In particular, it highlights the flexibility of the proposed procedure.

Rejection in regression is extremely rarely considered in the literature, an exception being [23] that views the reject option from a different perspective. There, the authors used the reject option from the side of $\varepsilon$-optimality, and therefore ensures that the prediction is inside a ball with radius $\varepsilon$ around the regression function with high probability. Their methodology is intrinsically associated with empirical risk minimization procedures. In contrast, our method is applicable to any estimation procedure. Closer related works to ours appears in classification with reject option literature [2, 4, 5, 6, 11, 13, 16, 22]. In particular, the present work can be viewed as an extension of the classification with reject option setting. Indeed, from a general perspective, the present contribution brings a deeper understanding of the reject option. Importantly, the conditional variance function appears to capture the main feature behind the abstention decision. In [6], the authors also provide rates of convergence for plug-in type approaches in the case of bounded rejection rate. However, their rates of convergence holds only under some margin type assumption [1, 17] and a smoothness assumption on the considered estimator. On the contrary, we do not require these assumptions to get valid rates of convergence.

## 2   Regression with reject option

In this section we introduce the regression with reject option setup and derive a general form of the optimal rule in this context. We additionally highlight the case of fixed rejection rate as our main framework. First of all, before we proceed, let us introduce some preliminary notation. Let $(X, Y)$ be a random couple taking its values in $\mathbb{R}^d \times \mathbb{R}$: here $X$ denotes a feature vector and $Y$ is the corresponding output. We denote by $\mathbb{P}$ the joint distribution of $(X, Y)$ and by $\mathbb{P}_X$ the marginal distribution of the feature $X$. Let $x \in \mathbb{R}^d$, we introduce the regression function $f^*(x) = \mathbb{E}\left[Y | X = x\right]$ as well as the conditional variance function $\sigma^2(x) = \mathbb{E}\left[(Y - f^*(X))^2 | X = x\right]$. We will give due attention to these two functions in our analysis. In addition, we denote by $\|\cdot\|$ the Euclidean on $\mathbb{R}^d$. Finally, $|\cdot|$ stands for the cardinality when dealing with a finite set.

### 2.1   Predictor with reject option

Let $f$ be some measurable real-valued function which must be viewed as a prediction function. A *predictor with reject option* $\Gamma_f$ associated to $f$ is defined as being any function that maps $\mathbb{R}^d$ onto $\mathcal{P}(\mathbb{R})$ such for all $x \in \mathbb{R}^d$, the output $\Gamma_f(x) \in \{\emptyset, \{f(x)\}\}$. We denote by $\Upsilon_f$ the set of all predictors with reject option that relies on $f$. Hence, in this framework, there are only two options for a particular $x \in \mathbb{R}^d$: whether the predictor with reject option outputs the empty set, meaning that no prediction is produced for $x$; or the output $\Gamma_f(x)$ is of size 1 and the prediction coincides with the value $f(x)$. The framework of regression with reject option naturally brings into play two important characteristics of a given predictor $\Gamma_f$. The first one is the rejection rate that we denote by $r(\Gamma_f) = \mathbb{P}(|\Gamma_f(X)| = 0)$ and the second one is the $L_2$ error when prediction is performed

$$\text{Err}(\Gamma_f) = \mathbb{E}\left[(Y - f(X))^2 \mid |\Gamma_f(X)| = 1\right] \ .$$

The ultimate goal in regression with reject option is to build a predictor $\Gamma_f$ with a small rejection rate that achieves a small conditional $L_2$ error as well. A natural way to make this happen is to embed these quantities into a measure of performance. To this end, let consider the following risk

$$\mathcal{R}_\lambda\left(\Gamma_f\right) = \mathbb{E}\left[(Y - f(X))^2 \mathbb{1}_{\{|\Gamma_f(X)|=1\}}\right] + \lambda\, r\left(\Gamma_f\right)\ ,$$

where $\lambda \geq 0$ is a tuning parameter which is responsible for compromising error and rejection rate: larger $\lambda$'s result in predictors $\Gamma_f$ with smaller rejection rates, but with larger errors. Hence, $\lambda$ can be interpreted as the price to pay for using the reject option. Note that the above risk $\mathcal{R}_\lambda$ has already been considered by [11] in the classification framework.

Minimizing the risk $\mathcal{R}_\lambda$, we derive an explicit expression of an optimal predictor with reject option.

**Proposition 2.1.** *Let $\lambda \geq 0$, and consider*

$$\Gamma_\lambda^* \in \arg\min \mathcal{R}_\lambda(\Gamma_f)\ ,$$

*where the minimum is taken over all predictors with rejection option $\Gamma_f \in \Upsilon_f$ and all measurable functions $f$. Then we have that*

1. *The optimal predictor with rejected option $\Gamma_\lambda^*$ can be written as*

$$\Gamma_\lambda^*(X) = \begin{cases} \{f^*(X)\} & \text{if } \sigma^2(X) \leq \lambda \\ \emptyset & \text{otherwise} \end{cases}. \tag{1}$$

2. *For any $\lambda < \lambda'$, the following holds*

$$\mathrm{Err}\left(\Gamma_\lambda^*\right) \leq \mathrm{Err}\left(\Gamma_{\lambda'}^*\right)\ \text{ and }\ r\left(\Gamma_\lambda^*\right) \geq r\left(\Gamma_{\lambda'}^*\right)\ .$$

Interestingly, this result shows that the oracle predictor relies on thresholding the conditional variance function $\sigma^2$. We believe that this is an important remark that provides an essential characteristic of the reject option in regression but also in classification. Indeed, it has been shown that the optimal classifier with reject option for classification is obtained by thresholding the function $f^*$ (see for instance [11]). However, in the binary case where $Y \in \{0, 1\}$, one has $\sigma^2(x) = f^*(x)(1 - f^*(x))$, and then thresholding $\sigma^2$ and $f^*$ are equivalent.

The second point of the proposition shows that the error and the rejection rate of the optimal predictor are working in two opposite directions *w.r.t.* $\lambda$ and then a compromise is required. We illustrate this aspect with the `airfoil` dataset, and the $k$NN predictor (see Section 5) in the contiguous Figure 1. The two curves correspond to the evaluation of the error $\mathrm{Err}(\hat{\Gamma}_\lambda)$ (blue-solid line) and the rejection rate $r(\hat{\Gamma}_\lambda)$ (red-dashed line) as a function of $\lambda$. In general any choice of the parameter $\lambda$ is difficult to interpret. Indeed, one of the major drawbacks of this approach is that any fixed $\lambda$ (or even an "optimal" value of this parameter) does not allow to control neither of the two parts of the risk function. Especially, the rejection rate can be arbitrary large.

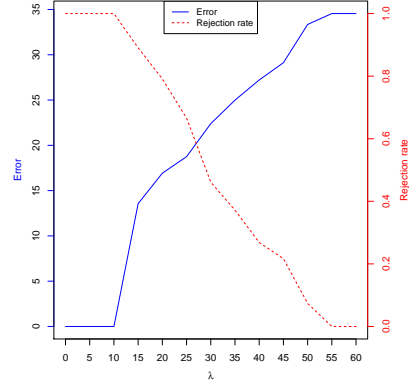

Figure 1: $\widehat{\mathrm{Err}}\left(\hat{\Gamma}_\lambda\right)$ and $\hat{r}\left(\hat{\Gamma}_\lambda\right)$ vs. $\lambda$.

For this reason, we investigate in Section 2.2 the setting where the rejection rate is fixed. We understand this rejection rate as a budget one has beforehand.

## 2.2 Optimal predictor with fixed rejection rate

In this section, we introduce the framework where the rejection rate is fixed or at least bounded. That is to say, for a given predictor with reject option $\Gamma_f$ and a given rejection rate $\varepsilon \in (0, 1)$, we ask that $\Gamma_f$ satisfies following constraint $r\left(\Gamma_f\right) \leq \varepsilon$. This kind of constraint has also been considered by [6] in the classification setting. Our objective becomes to solve the constrained problem[2]:

$$\Gamma_\varepsilon^* \in \arg\min\{\mathrm{Err}\left(\Gamma_f\right)\ :\ r\left(\Gamma_f\right) \leq \varepsilon\}\ . \tag{2}$$

In the same vein as Proposition 2.1, we aim at writing an explicit expression of $\Gamma_\varepsilon^*$, referred in what follows to as $\varepsilon$-predictor. However, this expression is not well identified in the general case. Therefore, we make the following mild assumption on the distribution of $\sigma^2(X)$, which translates the fact that the function $\sigma^2$ is not constant on any set with non-zero measure *w.r.t.* $\mathbb{P}_X$.

**Assumption 2.2.** *The cumulative distribution function $F_{\sigma^2}$ of $\sigma^2(X)$ is continuous.*

Let us denote by $F_{\sigma^2}^{-1}$ the generalized inverse of the cumulative distribution $F_{\sigma^2}$ defined for all $u \in (0,1)$ as $F_{\sigma^2}^{-1}(u) = \inf\{t \in \mathbb{R} \ : \ F_{\sigma^2}(t) \geq u\}$. Under Assumption 2.2 and from Proposition 2.1, we derive an explicit expression of the $\varepsilon$-predictor $\Gamma_\varepsilon^*$ given by (2).

**Proposition 2.3.** *Let $\varepsilon \in (0,1)$, and let $\lambda_\varepsilon = F_{\sigma^2}^{-1}(1-\varepsilon)$. Under Assumption 2.2, we have $\Gamma_\varepsilon^* = \Gamma_{\lambda_\varepsilon}^*$.*

As an immediate consequence of the above proposition and properties on quantile functions is that

$$r\left(\Gamma_\varepsilon^*\right) = \mathbb{P}\left(|\Gamma_\varepsilon^*(X)| = 0\right) = \mathbb{P}\left(\sigma^2(X) \geq \lambda_\varepsilon\right) = \mathbb{P}\left(F_{\sigma^2}(\sigma^2(X)) \geq 1-\varepsilon\right) = \varepsilon \ ,$$

and then the $\varepsilon$-predictor has rejection rate exactly $\varepsilon$. The continuity Assumption 2.2 is a sufficient condition to ensure that this property holds true. Besides, from this assumption, the $\varepsilon$-predictor can be expressed as follows

$$\Gamma_\varepsilon^*(x) = \begin{cases} \{f^*(x)\} & \text{if} \ \ F_{\sigma^2}(\sigma^2(x)) \leq 1-\varepsilon \\ \emptyset & \text{otherwise} \end{cases} \ . \tag{3}$$

Finally, as suggested by Proposition 2.1 and 2.3, the performance of a given predictor with reject option $\Gamma_f$ is measured through the risk $\mathcal{R}_\lambda$ when $\lambda = \lambda_\varepsilon$. Then, its excess risk is given by

$$\mathcal{E}_{\lambda_\varepsilon}\left(\Gamma_f\right) = \mathcal{R}_{\lambda_\varepsilon}(\Gamma_f) - \mathcal{R}_{\lambda_\varepsilon}(\Gamma_\varepsilon^*) \ ,$$

for which the following result provides a closed formula.

**Proposition 2.4.** *Let $\varepsilon \in (0,1)$. For any predictor $\Gamma_f$, we have*

$$\mathcal{E}_{\lambda_\varepsilon}\left(\Gamma_f\right) = \mathbb{E}_X\left[(f^*(X) - f(X))^2 \mathbb{1}_{\{|\Gamma_f(X)|=1\}}\right] + \mathbb{E}_X\left[|\sigma^2(X) - \lambda_\varepsilon| \mathbb{1}_{\{|\Gamma_f(X)| \neq |\Gamma_\varepsilon^*(X)|\}}\right] \ .$$

The above excess risk consists of two terms that translates two different aspect of the regression with reject option problem. The first one is related to the $L_2$ risk of the prediction function $f$ and is rather classical in the regression setting. On contrast, the second is related to the reject option problem. It is dictated by the behavior of the conditional variance $\sigma^2$ around the threshold $\lambda_\varepsilon$.

## 3 Plug-in $\varepsilon$-predictor with reject option

We devote this section to the study of a data-driven predictor with reject option based on the *plug-in* principle that mimics the optimal rule derived in Proposition 2.3.

### 3.1 Estimation strategy

Equation (3) indicates that a possible way to estimate $\Gamma_\varepsilon^*$ relies on the plug-in principle. To be more specific, Eq. (3) suggests that estimating $f^*$ and $\sigma^2$, as well as the cumulative distribution $F_{\sigma^2}$ would be enough to get an estimator of $\Gamma_\varepsilon^*$. To build such predictor, we first introduce a learning sample $\mathcal{D}_n = \{(X_i, Y_i), \ i = 1, \dots, n\}$ which consists of $n$ independent copies of $(X, Y)$. This dataset helps us to construct estimators $\hat{f}$ and $\hat{\sigma}^2$ of the regression function $f^*$ and the conditional variance function $\sigma^2$ respectively. In this paper, we focus on estimator $\hat{\sigma}^2$ which relies on the residual-based methods [9]. Based on $\mathcal{D}_n$, the estimator $\hat{\sigma}^2$ is obtained by solving the regression problem of the output variable $(Y - \hat{f}(X))^2$ on the input variable $X$. Estimating the last quantity $F_{\sigma^2}$ is rather simple by replacing cumulative distribution function by its empirical version. Since this term only depends on the marginal distribution $\mathbb{P}_X$, we estimate it using a second *unlabeled* dataset $\mathcal{D}_N = \{X_{n+1}, \dots, X_{n+N}\}$ composed of $N$ independent copies of $X$. This is an important feature of our methodology since unlabeled data are usually easy to get. The dataset $\mathcal{D}_N$ is assumed to be independent of $\mathcal{D}_n$. We set

$$\hat{F}_{\hat{\sigma}^2}(\cdot) = \frac{1}{N}\sum_{i=1}^{N} \mathbb{1}_{\{\hat{\sigma}^2(X_{n+i}) \leq \cdot\}} \ ,$$

as an estimator for $F_{\sigma^2}$. With this notation, the *plug-in $\varepsilon$-predictor* is the predictor with reject option defined for each $x \in \mathbb{R}^d$ as

$$\hat{\Gamma}_\varepsilon(x) = \begin{cases} \left\{ \hat{f}(x) \right\} & \text{if } \hat{F}_{\hat{\sigma}^2}(\hat{\sigma}^2(x)) \leq 1 - \varepsilon \\ \emptyset & \text{otherwise} \end{cases} \tag{4}$$

It is worth noting that the proposed methodology is flexible enough to rely upon any off-the-shelf estimators of the regression function $f^*$ and the conditional variance function $\sigma^2$.

## 3.2 Consistency of plug-in $\varepsilon$-predictors

In this part, we investigate the statistical properties of the plug-in $\varepsilon$-predictors with reject option. This analysis requires an additional assumption on the following quantity

$$F_{\hat{\sigma}^2}(\cdot) = \mathbb{P}_X \left( \hat{\sigma}^2(X) \leq \cdot | \mathcal{D}_n \right) .$$

**Assumption 3.1.** *The cumulative distribution function $F_{\hat{\sigma}^2}$ of $\hat{\sigma}^2(X)$ is continuous.*

This condition is analogous to Assumption 2.2 but deals with the estimator $\hat{\sigma}^2(X)$ instead of the true conditional variance $\sigma^2(X)$. This difference makes Assumption 3.1 rather weak as the estimator $\hat{\sigma}^2(X)$ is chosen by the practitioner. Moreover, we can make any estimator satisfy this condition by providing a smoothed version of it. We illustrate this strategy with $k$NN algorithm in Section 4. Next theorem is the main result of this section, it establishes the consistency of the predictor $\hat{\Gamma}_\varepsilon$ to the optimal one.

**Theorem 3.2.** *Let $\varepsilon \in (0, 1)$. Assume that $\sigma^2$ is bounded, $\hat{f}$ is a consistent estimator of $f^*$ w.r.t. the $L_2$ risk, and $\hat{\sigma}^2$ is a consistent estimator of $\sigma^2$ w.r.t. the $L_1$ risk. Under Assumptions 2.2- 3.1, the followings hold*

$$\mathbb{E}\left[ \mathcal{E}_{\lambda_\varepsilon}\left( \hat{\Gamma}_\varepsilon \right) \right] \xrightarrow[n,N \to +\infty]{} 0, \quad and \quad \mathbb{E}\left[ |r(\hat{\Gamma}_\varepsilon) - \varepsilon| \right] \leq C N^{-1/2} ,$$

*where $C > 0$ is an absolute constant.*

This theorem establishes the fact that the plug-in $\varepsilon$-predictor behaves asymptotically as well as the optimal $\varepsilon$-predictor both in terms of risk and rejection rate. The convergence of the rejection rate requires only Assumption 3.1 which is rather weak and can even be removed following the process detailed in Section 4.2. In particular, the theorem shows that the rejection rate of the plug-in $\varepsilon$-predictor is of level $\varepsilon$ up to a term of order $O(N^{-1/2})$. This rate is similar to the one obtained in the classification setting [6]. It relies on the difference between the cumulative distribution $F_{\hat{\sigma}^2}$ and its empirical counterpart $\hat{F}_{\hat{\sigma}^2}$ that is controlled using Dvoretzky-Kiefer-Wolfowitz Inequality [15]. Interestingly, this result applies to any consistent estimators of $f^*$ and $\sigma^2$.

The estimation of regression function $f^*$ is widely studied and suitable algorithm such as random forests, kernel procedures, or $k$NN estimators can be used, see [3, 8, 18, 20, 21]. The estimation of the conditional variance function which relies on the residual-based methods has also been extensively studied based on kernel procedures, see for instance [7, 9, 10, 12, 19]. In the next section, we derive rates of convergence in the case where both estimators $\hat{f}$ and $\hat{\sigma}^2$ rely on the $k$NN algorithm. In particular, we establish rates of convergence for $\hat{\sigma}^2$ in sup norm (see Proposition D.5 in the supplementary material).

## 4 Application to $k$NN algorithm: rates of convergence

The plug-in $\varepsilon$-predictor $\hat{\Gamma}_\varepsilon$ relies on estimators of the regression and the conditional variance functions. In this section, we consider the specific case of $k$NN based estimations. We refer to the resulting predictor as *$k$NN predictor with reject option*. Specifically, we establish rates of convergence for this procedure. In addition, since $k$NN estimator of $\sigma^2$ violates Assumption 3.1, applying our methodology to $k$NN has the benefit of illustrating the smoothing technique to make this condition be satisfied.

## 4.1 Assumptions

To study the performance of the $k$NN predictor with reject option in the finite sample regime, we assume that $X$ belongs to a regular compact set $\mathcal{C} \subset \mathbb{R}^d$, see [1]. Besides, we make the following assumptions.

**Assumption 4.1.** *The functions $f^*$ and $\sigma^2$ are Lipschitz.*

**Assumption 4.2** (Strong density assumption)**.** *We assume that the marginal distribution $\mathbb{P}_X$ admits a density $\mu$ w.r.t to the Lebesgue measure such that for all $x \in \mathcal{C}$, we have $0 < \mu_{\min} \leq \mu(x) \leq \mu_{\max}$.*

These two assumptions are rather classical when we deal with rate of convergence and we refer the reader to the baseline books [8, 21]. In particular, we point out that the strong density assumption has been introduced in the context of binary classification for instance in [1]. The last assumption that we require highlights the behavior of $\sigma^2$ around the threshold $\lambda_\varepsilon$.

**Assumption 4.3** ($\alpha$-exponent assumption)**.** *We say that $\sigma^2$ has exponent $\alpha \geq 0$ (at level $\lambda_\varepsilon$) with respect to $\mathbb{P}_X$ if there exists $c^* > 0$ such that for all $t > 0$*

$$\mathbb{P}_X \left( 0 < |\sigma^2(X) - \lambda_\varepsilon| \leq t \right) \leq c^* t^\alpha .$$

This assumption has been first introduced in [17] and is also referred as Margin assumption in the binary classification setting, see [14]. For $\alpha > 0$, Assumption 4.3 ensures that the random variable $\sigma^2(X)$ can not concentrate too much around the threshold $\lambda_\varepsilon$. It allows to derive faster rates of convergence. Note that, if $\alpha = 0$ there is no assumption.

## 4.2 $k$NN predictor with reject option

For any $x \in \mathbb{R}^d$, we denote by $(X_{(i,n)}(x), Y_{(i,n)}(x)), i = 1, \ldots n$ the reordered data according to the $\ell_2$ distance in $\mathbb{R}^d$, meaning that $\|X_{(i,n)}(x) - x\| < \|X_{(j,n)}(x) - x\|$ for all $i < j$ in $\{1, \ldots, n\}$. Note that Assumption 4.2 ensures that ties occur with probability 0 (see [8] for more details). Let $k = k_n$ be an integer. The $k$NN estimator of $f^*$ and $\sigma^2$ are then defined, for all $x$, as follows

$$\hat{f}(x) = \frac{1}{k_n} \sum_{i=1}^{k_n} Y_{(i,n)}(x) \ \text{ and } \ \hat{\sigma}^2(x) = \frac{1}{k_n} \sum_{i=1}^{k_n} \left( Y_{(i,n)}(x) - \hat{f}(X_{(i,n)}(x)) \right)^2 .$$

Conditional on $\mathcal{D}_n$, the cumulative distribution function $F_{\hat{\sigma}^2}$ is not continuous and then Assumption 3.1 does not hold. To avoid this issue, we introduce a random perturbation $\zeta$ distributed according to the Uniform distribution on $[0, u]$ that is independent from every other random variable where $u > 0$ is a (small) fixed real number that will be specified later. Then, we define the random variable $\bar{\sigma}^2(X, \zeta) := \hat{\sigma}^2(X) + \zeta$. It is not difficult to see that, conditional on $\mathcal{D}_n$ the cumulative distribution $F_{\bar{\sigma}^2}$ of $\bar{\sigma}^2(X, \zeta)$ is continuous. Furthermore, by the triangle inequality, the consistency of $\hat{\sigma}^2$ implies the consistency of $\bar{\sigma}^2$ provided that $u$ tends to 0. Therefore, we naturally define the $k$NN predictor with reject option as follows.

Let $(\zeta_1, \ldots, \zeta_N)$ be independent copies of $\zeta$ and independent of every other random variable. We set

$$\hat{F}_{\bar{\sigma}^2}(.) = \frac{1}{N} \sum_{i=1}^{N} \mathbb{1}_{\{\hat{\sigma}^2(X_{n+i}) + \zeta_i \leq \cdot\}} ,$$

and the $k$NN $\varepsilon$-predictor with reject option is then defined for all $x$ and $\zeta$ as

$$\hat{\Gamma}_\varepsilon(x, \zeta) = \begin{cases} \left\{ \hat{f}(x) \right\} & \text{if } \hat{F}_{\bar{\sigma}^2}(\bar{\sigma}^2(x, \zeta)) \leq 1 - \varepsilon \\ \emptyset & \text{otherwise} . \end{cases}$$

## 4.3 Rates of convergence

In this section, we derive the rates of convergence of the $k$NN $\varepsilon$-predictor in the following framework. We assume that $Y$ is bounded or that $Y$ satisfies

$$Y = f^*(X) + \sigma(X)\xi , \tag{5}$$

where $\xi$ is independent of $X$ and distributed according to a standard normal distribution. Note that these assumptions covers a broad class of applications. Under these assumptions, we can state the following result.

**Theorem 4.4.** *Grant Assumptions 2.2, 4.1, 4.2, and 4.3. Let $\varepsilon \in (0,1)$, if $k_n \propto n^{2/(d+2)}$, and $u \leq n^{-1/(d+2)}$, then the kNN $\varepsilon$-predictor $\hat{\Gamma}_\varepsilon$ satisfies*

$$\mathbb{E}\left[\mathcal{E}_{\lambda_\varepsilon}\left(\hat{\Gamma}_\varepsilon\right)\right] \leq C\left(n^{-2/(d+2)} + \log(n)^{(\alpha+1)}n^{-(\alpha+1)/(d+2)} + N^{-1/2}\right) \ ,$$

*where $C > 0$ is a constant which depends on $f^*$, $\sigma^2$, $c_0$, $c^*$, $\alpha$, $\mathcal{C}$, and on the dimension d.*

Each part of the above rate describes a given feature of the problem. The first one relies on the estimation error of the regression function. The second one, which depends in part on the parameter $\alpha$ from Assumption 4.3, is due to the estimation error in sup norm of the conditional variance $\mathbb{E}\left[\left(\sup_{x\in\mathcal{C}}\left|\hat{\sigma}^2(x) - \sigma^2(x)\right|\right)\right] \leq C\log(n)n^{-1/(d+2)}$ stated in Proposition D.5 in the supplementary material. Notice that for $\alpha = 1$, the second term is of the same order (up to logarithmic factor) as the term corresponding to the estimation of the regression function. The last term is directly linked to the estimation of the threshold $\lambda_\varepsilon$. Lastly, for $\alpha > 1$, we observe, provided that the size of the unlabeled sample $N$ is sufficiently large, that this rate is the same as the rate of $\hat{f}$ in $L_2$ norm which is then the best situation that we can expect for the rejection setting.

## 5 Numerical experiments

In this section, we present numerical experiments to illustrate the performance of the plug-in $\varepsilon$-predictor. The construction process of this predictor is described in Section 3.1 and relies on estimators of the regression and the conditional variance functions. The code used for the implementation of the plug-in $\varepsilon$-predictor can be found at `https://github.com/ZaouiAmed/Neurips2020_RejectOption`. For this experimental study, we consider the same algorithm for both estimation tasks and build three plug-in $\varepsilon$-predictors based respectively on support vector machines (`svm`), random forests (`rf`), and $k$NN (`knn`) algorithms. Besides, to avoid non continuity issues, we add the random perturbation $\zeta \sim \mathcal{U}[0, 10^{-10}]$ to all of the considered methods as described in Section 4.2. The performance is evaluated on two benchmark datasets: *QSAR aquatic toxicity* and *Airfoil Self-Noise* coming from the UCI database. We refer to these two datasets as `aquatic` and `airfoil` respectively. For all datasets, we split the data into three parts (50 % train labeled, 20 % train unlabeled, 30 % test). The first part is used to estimate both regression and variance functions, while the second part is used to compute the empirical cumulative distribution function. Finally, for each $\varepsilon \in \{i/10, \ i = 0, \ldots, 9\}$ and each plug-in $\varepsilon$-predictor, we compute the empirical rejection rate $\hat{r}$ and the empirical error $\widehat{\text{Err}}$ on the test set. This procedure is repeated 100 times and we report the average performance on the test set alongside its standard deviation. We employ the 10-fold cross-validation to select the parameter $k \in \{5, 10, 15, 20, 30, 50, 70, 100, 150\}$ of the $k$NN algorithm. For random forests and svm procedures, we used respectively the R packages `randomForest` and `e1071` with default parameters.

### 5.1 Datasets

The datasets used for the experiments are briefly described bellow:
*QSAR aquatic toxicity* has been used to develop quantitative regression QSAR models to predict acute aquatic toxicity towards the fish Pimephales promelas. This dataset is composed of $n = 546$ observations for which 8 numerical features are measured. The output takes its values in $[0.12, 10.05]$.
*Airfoil Self-Noise* is composed of $n = 1503$ observations for which 5 features are measured. This dataset is obtained from a series of aerodynamic and acoustic tests. The output is the scaled sound pressure level, in decibels. It takes its values in $[103, 140]$.

Since the variance function plays a key role in the construction of the plug-in $\varepsilon$-predictor, we display in Figure 2 the histogram of an estimate of $\sigma^2(X)$ produced by the random forest algorithm. More specifically, for each $i = 1, \ldots, n$, we evaluate $\hat{\sigma}^2(X_i)$ by 10-fold cross-validation and build the histogram of $(\hat{\sigma}^2(X_i))_{i=1,\ldots,n}$ thereafter. Left and right panels of Figure 2 deal respectively with the `aquatic` and `airfoil` datasets and reflect two different situations where the use of reject option is relevant. The estimated variance in the `airfoil` dataset is typically large (about $40\%$ of the values are larger than 10) and then we may have some doubts in the associated prediction. According to the `aquatic` dataset, main part of the estimated values $\hat{\sigma}^2$ is smaller than 1 and then the use of the reject option may seem less significant. However, in this case, the predictions produced by the plug-in $\varepsilon$-predictors would be very accurate.

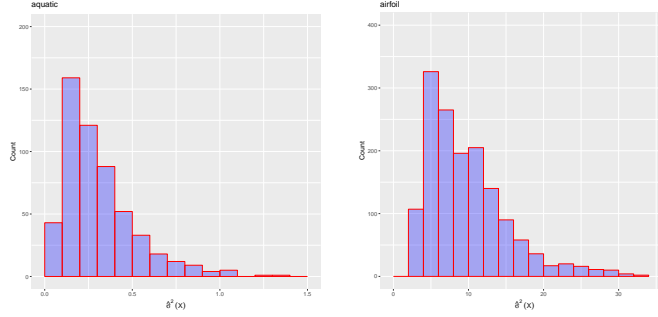

Figure 2: Histogram of the estimates of $\sigma^2(X)$

## 5.2 Results

We present the obtained results in Figure 3 and Table 1. We make a focus on the values of $\varepsilon \in \{0, 0.2, 0.5, 0.8\}$. As a general picture, the results are reflecting our theory: the empirical errors of the plug-in $\varepsilon$-predictors are decreasing *w.r.t.* $\varepsilon$ for both datasets and their empirical rejection rates are very close to their expected values. Indeed, Table 1 displays how precise is the estimation of the rejection rate whatever the method used. This is in accordance with our theoretical findings. Moreover, the empirical errors of the plug-in $\varepsilon$-predictors based on the random forests and $k$NN algorithms are decreasing *w.r.t.* $\varepsilon$ for both datasets. As expected, the use of the reject option improves the prediction precision. As an illustration, for `airfoil` dataset and the predictor based on random forests, the error is divided by 2 if we reject $50\%$ of the data. However, we discover that the decrease for the prediction error is not systematic. In the case of plug-in $\varepsilon$-predictor based on the svm algorithm and with the `aquatic` dataset, we observe a strange curve for the error rate (see Figure 3-left). We conjecture that this phenomenon is due to a poor estimation of the variance. Indeed, in Figure 4, we present the performance of some kind of hybrid plug-in $\varepsilon$-predictors: we still use the svm algorithm to estimate the regression function; the variance function estimation is done based on svm (dashed line), random forests (dotted line), and $k$NN (dash-dotted line). From Figure 4, we observe that the empirical error $\widehat{\text{Err}}$ is now decreasing *w.r.t.* $\varepsilon$ for the hybrid predictors based on svm and random forests, and that the performance is quite good.

Table 1: Performances of the three plug-in $\varepsilon$-predictors on the real datasets `aquatic`, and `airfoil`.

| | aquatic | | | | | | airfoil | | | | | |
| | svm | | rf | | knn | | svm | | rf | | knn | |
| 1-$\varepsilon$ | $\widehat{\text{Err}}$ | $1 - \hat{r}$ | $\widehat{\text{Err}}$ | $1 - \hat{r}$ | $\widehat{\text{Err}}$ | $1 - \hat{r}$ | $\widehat{\text{Err}}$ | $1 - \hat{r}$ | $\widehat{\text{Err}}$ | $1 - \hat{r}$ | $\widehat{\text{Err}}$ | $1 - \hat{r}$ |
|---|---|---|---|---|---|---|---|---|---|---|---|---|
| 1 | 1.38 (0.18) | 1.00 (0.00) | 1.34 (0.18) | 1.00 (0.00) | 2.29 (0.27) | 1.00 (0.00) | 11.81 (1.03) | 1.00 (0.00) | 14.40 (1.04) | 1.00 (0.00) | 35.40 (2.05) | 1.00 (0.00) |
| 0.8 | 1.08 (0.17) | 0.81 (0.05) | 1.04 (0.16) | 0.80 (0.05) | 1.98 (0.26) | 0.80 (0.04) | 8.27 (0.86) | 0.80 (0.03) | 10.26 (0.95) | 0.80 (0.03) | 31.13 (1.96) | 0.80 ( 0.03) |
| 0.5 | 0.91 (0.18) | 0.50 (0.06) | 0.81 (0.18) | 0.50 (0.06) | 1.51 (0.30) | 0.50 (0.06) | 5.15 (0.92) | 0.50 (0.04) | 7.22 (0.92) | 0.50 (0.3) | 22.42 (2.13) | 0.50 (0.03) |
| 0.2 | 1.01 (0.32) | 0.19 (0.05) | 0.55 (0.21) | 0.20 (0.05) | 0.75 (0.37) | 0.19 (0.05) | 2.6 (0.64) | 0.20 (0.03) | 4.00 (0.74) | 0.20 (0.03) | 17.27 (3.00) | 0.19 (0.03) |

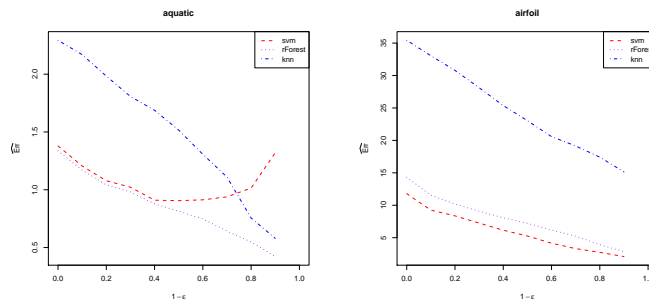

Figure 3: Visual description of the performance of three plug-in $\varepsilon$-predictors on the `aquatic`, and `airfoil` datasets.

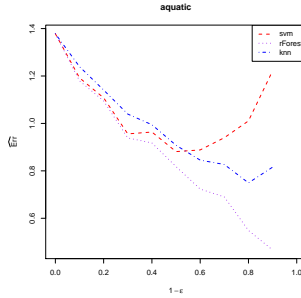

Figure 4: Additional description of the performance of the plug-in procedure on `aquatic` dataset.

## 6 Conclusion

We generalized the use of the reject option to the regression setting. We investigated the particular setting where the rejection rate is bounded. In this framework, an optimal rule is derived, it relies on thresholding of the variance function. Based on the *plug-in* principle, we derived a semi-supervised algorithm that can be applied on top of any off-the-shelf estimators of both regression and variance functions. One of the main features of the proposed procedure is that it precisely controls the probability of rejection. We derived general consistency results on rejection rate and on excess risk. We also established rates of convergence for the predictor with reject option when the regression and the variance functions are estimated by $k$NN algorithm. In future work, we plan to apply our methodology to the high-dimensional setting, taking advantage of sparsity structure of the data.

## Broader impact

Approaches based on reject option may be helpful at least from two perspectives. First, when human action is limited by time or any other constraint, reject option is an efficient tool to prioritize the human action. On the other hand, in a world where automatic decisions need to be balanced and considered with caution, abstaining from prediction is one way to prevent from damageable decision-making. In particular, human is more likely able to detect anomalies such as bias in data. In a manner of speaking, the use of the reject option compromises between human and machines! Our numerical and theoretical analyses support this idea, in particular because our estimation of the rejection rate is accurate.

While the rejection rate has to be fixed according to the considered problem, it appears that the main drawback of our approach is that border instances may be automatically treated while they would have deserved a human consideration. From a general perspective, this is a weakness of all methods based on reject option. This inconvenience is even stronger when the conditional variance function is poorly estimated.

## Footnotes

[2]By abuse of notation, we refer to $\Gamma_\lambda^*$ as the solution of the penalized problem and to $\Gamma_\varepsilon^*$ as the solution of the constrained problem.

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
