[Supplementary Material]

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

[3]The only difference between $\tilde{\Gamma}_\varepsilon(x, \zeta)$ and $\tilde{\Gamma}_\varepsilon(x)$ given in (8) is the dependency in $\zeta$ that is hidden inside $\bar{\sigma}^2$. To avoid useless additional notation, we write $\tilde{\Gamma}_\varepsilon$ for both pseudo-oracles.

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

## Supplementary material

This supplementary material is organized as follows. Section A provides all proofs of results related to the optimal predictors (that is, Propositions 2.1, 2.3 2.4). In Sections B and C we prove Theorem 3.2 that establishes the consistency and Theorem 4.4 that states the rates of convergence of the plug-in $\varepsilon$-predictor $\hat{\Gamma}_\varepsilon$ respectively. We further establish several finite sample guarantees on $k$NN estimator in Section D. To help readability of the paper, we provide in Section E some technical tools that are used for the proofs.

## A  Proofs for optimal predictors

*Proof of Proposition 2.1.* By definition of $\mathcal{R}_\lambda$, we have for any predictor with reject option $\Gamma_f$

$$
\begin{aligned}
\mathcal{R}_\lambda\left(\Gamma_f\right) &= \mathbb{E}\left[(Y - f(X))^2 \mathbb{1}_{\{|\Gamma_f(X)|=1\}}\right] + \lambda \mathbb{P}(|\Gamma_f(X)| = 0) \\
&= \mathbb{E}\left[(Y - f^*(X) + f^*(X) - f(X))^2 \mathbb{1}_{\{|\Gamma_f(X)|=1\}}\right] + \lambda \mathbb{P}(|\Gamma_f(X)| = 0) \\
&= \mathbb{E}\left[(Y - f^*(X))^2 \mathbb{1}_{\{|\Gamma_f(X)|=1\}}\right] + \mathbb{E}\left[(f^*(X) - f(X))^2 \mathbb{1}_{\{|\Gamma_f(X)|=1\}}\right] \\
&\quad + 2\mathbb{E}\left[(Y - f^*(X))(f^*(X) - f(X))\mathbb{1}_{\{|\Gamma_f(X)|=1\}}\right] + \lambda \mathbb{P}(|\Gamma_f(X)| = 0) \ .
\end{aligned}
$$

Since

$$
\mathbb{E}\left[(Y - f^*(X))(f^*(X) - f(X))\mathbb{1}_{\{|\Gamma_f(X)|=1\}}\right] = 0 \ ,
$$

and

$$
\mathbb{E}\left[(Y - f^*(X))^2 \mathbb{1}_{\{|\Gamma_f(X)|=1\}}\right] = \mathbb{E}\left[\sigma^2(X)\mathbb{1}_{\{|\Gamma_f(X)|=1\}}\right] \ ,
$$

we deduce,

$$
\begin{aligned}
\mathcal{R}_\lambda(\Gamma_f) &= \mathbb{E}\left[(f^*(X) - f(X))^2 \mathbb{1}_{\{|\Gamma_f(X)|=1\}}\right] + \mathbb{E}\left[\sigma^2(X)\mathbb{1}_{\{|\Gamma_f(X)|=1\}} + \lambda(1 - \mathbb{1}_{\{|\Gamma_f(X)|=1\}})\right] \\
&= \mathbb{E}\left[\left\{(f^*(X) - f(X))^2 + (\sigma^2(X) - \lambda)\right\} \mathbb{1}_{\{|\Gamma_f(X)|=1\}}\right] + \lambda \ . \tag{6}
\end{aligned}
$$

Clearly, on the event $\{|\Gamma_f(X)| = 1\}$, the mapping $f \mapsto (f^*(X) - f(X))^2 + (\sigma^2(X) - \lambda)$ achieves its minimum at $f = f^*$. Then, it remains to consider the minimization of

$$
\Gamma \mapsto \mathbb{E}\left[\left\{(\sigma^2(X) - \lambda)\right\} \mathbb{1}_{\{|\Gamma(X)|=1\}}\right] + \lambda \ ,
$$

on the set $\Upsilon_{f^*}$, which leads to $\{|\Gamma(X)| = 1\} = \{\sigma^2(X) \leq \lambda\}$. Putting all together, we get

$$
\{|\Gamma_\lambda^*(X)| = 1\} = \{\sigma^2(X) \leq \lambda\} \quad \text{and on this event } \Gamma_\lambda^*(X) = \{f^*(X)\} \ ,
$$

and point 1. of Proposition 2.1 is proven. For the second point, we observe that for $0 < \lambda < \lambda'$,

$$
\{|\Gamma_\lambda^*(X)| = 1\} = \{\sigma^2(X) \leq \lambda\} \subset \{\sigma^2(X) \leq \lambda'\} = \{|\Gamma_{\lambda'}^*(X)| = 1\} \ .
$$

From this inclusion, we deduce $r(\Gamma_{\lambda'}^*) \leq r(\Gamma_\lambda^*)$. Furthermore, using the relation $\{|\Gamma_\lambda^*(X)| = 1\} = \{\sigma^2(X) \leq \lambda\}$ and if we denote by $a_\lambda = \mathbb{P}\left(|\Gamma_\lambda^*(X)| = 1\right)$ we have

$$
\begin{aligned}
\mathrm{Err}\left(\Gamma_\lambda^*\right) - \mathrm{Err}\left(\Gamma_{\lambda'}^*\right) &= \frac{1}{a_\lambda}\mathbb{E}\left[(Y - f^*(X))^2 \mathbb{1}_{\{\sigma^2(X)\leq\lambda\}}\right] - \frac{1}{a_{\lambda'}}\mathbb{E}\left[(Y - f^*(X))^2 \mathbb{1}_{\{\sigma^2(X)\leq\lambda'\}}\right] \\
&= \left(\frac{1}{a_\lambda} - \frac{1}{a_{\lambda'}}\right)\mathbb{E}\left[(Y - f^*(X))^2 \mathbb{1}_{\{\sigma^2(X)\leq\lambda\}}\right] \\
&\qquad - \frac{1}{a_{\lambda'}}\mathbb{E}\left[(Y - f^*(X))^2 \mathbb{1}_{\{\lambda<\sigma^2(X)\leq\lambda'\}}\right] \ . \tag{7}
\end{aligned}
$$

By definition of $\sigma^2(X)$, we can write

$$
\begin{aligned}
\mathbb{E}\left[(Y - f^*(X))^2 \mathbb{1}_{\{\sigma^2(X)\leq\lambda\}}\right] &= \mathbb{E}\left[\mathbb{1}_{\{\sigma^2(X)\leq\lambda\}}\mathbb{E}\left[(Y - f^*(X))^2|X\right]\right] \\
&= \mathbb{E}\left[\mathbb{1}_{\{\sigma^2(X)\leq\lambda\}}\sigma^2(X)\right] \leq \lambda a_\lambda,
\end{aligned}
$$

and then

$$
\left(\frac{1}{a_\lambda} - \frac{1}{a_{\lambda'}}\right)\mathbb{E}\left[(Y - f^*(X))^2 \mathbb{1}_{\{\sigma^2(X)\leq\lambda\}}\right] \leq \lambda\left(1 - \frac{a_\lambda}{a_{\lambda'}}\right) \ .
$$

In the same way, we obtain

$$
\frac{1}{a_{\lambda'}}\mathbb{E}\left[(Y - f^*(X))^2 \mathbb{1}_{\{\lambda\leq\sigma^2(X)\leq\lambda'\}}\right] \geq \frac{\lambda}{a_{\lambda'}}\left(a_{\lambda'} - a_\lambda\right) = \lambda\left(1 - \frac{a_\lambda}{a_{\lambda'}}\right) \ .
$$

From Equation (7), we then get $\mathrm{Err}\left(\Gamma_\lambda^*\right) \leq \mathrm{Err}\left(\Gamma_{\lambda'}^*\right)$. □

*Proof of Proposition 2.3.* First of all, observe that for any $\varepsilon \in (0, 1)$, if we set $\lambda_\varepsilon = F_{\sigma^2}^{-1}(1 - \varepsilon)$, then the optimal predictor $\Gamma_\lambda^*$ given by (1) with $\lambda = \lambda_\varepsilon$ is such that,

$$r\left(\Gamma_{\lambda_\varepsilon}^*\right) = \mathbb{P}\left(\sigma^2(X) \geq \lambda_\varepsilon\right) = \mathbb{P}\left(F_{\sigma^2}(\sigma^2(X)) \geq 1 - \varepsilon\right) = \varepsilon .$$

We need to prove that any predictor $\Gamma_f$ such that $r(\Gamma_f) = \varepsilon'$ with $\varepsilon' \leq \varepsilon$, satisfies $\mathrm{Err}(\Gamma_f) \geq \mathrm{Err}\left(\Gamma_{\lambda_\varepsilon}^*\right)$. To this end, consider $\Gamma_{\lambda_{\varepsilon'}}^*$ with $\lambda_{\varepsilon'} = F_{\sigma^2}^{-1}(1 - \varepsilon')$. On one hand, by optimality of $\Gamma_{\lambda_{\varepsilon'}}^*$ (*cf.* point 1. of Proposition 2.1), we have

$$\mathrm{Err}(\Gamma_f) - \mathrm{Err}\left(\Gamma_{\lambda_{\varepsilon'}}^*\right) = \frac{1}{1 - \varepsilon'}\left(\mathcal{R}_{\lambda_{\varepsilon'}}(\Gamma_f) - \mathcal{R}_{\lambda_{\varepsilon'}}\left(\Gamma_{\lambda_{\varepsilon'}}^*\right)\right) \geq 0 .$$

On the other hand, since $\varepsilon' \leq \varepsilon$ implies $\lambda_\varepsilon \leq \lambda_{\varepsilon'}$, point 2. of Proposition 2.1 reads as

$$\mathrm{Err}\left(\Gamma_{\lambda_\varepsilon}^*\right) \leq \mathrm{Err}\left(\Gamma_{\lambda_{\varepsilon'}}^*\right) .$$

Combining these two facts gives the desired result. $\qquad\square$

*Proof of Proposition 2.4.* First, from Equation (6), we have the following decomposition

$$
\begin{aligned}
\mathcal{R}_{\lambda_\varepsilon}(\Gamma_f) &= \mathbb{E}\left[\left\{(f^*(X) - f(X))^2 + \sigma^2(X) - \lambda_\varepsilon\right\}\mathbb{1}_{\{|\Gamma_f(X)|=1\}}\right] + \lambda_\varepsilon \\
&= \mathbb{E}\left[(f^*(X) - f(X))^2\mathbb{1}_{\{|\Gamma_f(X)|=1\}}\right] + \mathbb{E}\left[(\sigma^2(X) - \lambda_\varepsilon)\mathbb{1}_{\{|\Gamma_f(X)|=1\}}\right] + \lambda_\varepsilon .
\end{aligned}
$$

Therefore, we deduce

$$\mathcal{E}(\Gamma_f) = \mathbb{E}\left[(f^*(X) - f(X))^2\mathbb{1}_{\{|\Gamma_f(X)|=1\}}\right] + \mathbb{E}\left[(\sigma^2(X) - \lambda_\varepsilon)\left\{\mathbb{1}_{\{|\Gamma_f(X)|=1\}} - \mathbb{1}_{\{|\Gamma_\varepsilon^*(X)|=1\}}\right\}\right],$$

and the result follows from the fact that all non zero values of $\mathbb{1}_{\{|\Gamma_f(X)|=1\}} - \mathbb{1}_{\{|\Gamma_\varepsilon^*(X)|=1\}}$ equal the sign of $\left(\sigma^2(X) - \lambda_\varepsilon\right)$ due to the fact that $\{|\Gamma_\varepsilon^*(X)| = 1\} = \{\sigma^2(X) - \lambda_\varepsilon \leq 0\}$. $\qquad\square$

# B  Proof of the consistency results: Theorem 3.2

The consistency of $\hat{\Gamma}_\varepsilon$ consists in the introduction of a pseudo oracle $\varepsilon$-predictor $\tilde{\Gamma}_\varepsilon$ defined for all $x \in \mathbb{R}^d$ by

$$\tilde{\Gamma}_\varepsilon(x) = \begin{cases} \left\{\hat{f}(x)\right\} & \text{if } \hat{\sigma}^2(x) \leq F_{\hat{\sigma}^2}^{-1}(1 - \varepsilon) \\ \emptyset & \text{otherwise} . \end{cases} \tag{8}$$

This predictor differs from $\hat{\Gamma}_\varepsilon$ in that it knows the marginal distribution $\mathbb{P}_X$ and then it has rejection rate exactly $\varepsilon$. Then, we consider the following decomposition

$$\mathbb{E}\left[\mathcal{E}_{\lambda_\varepsilon}\left(\hat{\Gamma}_\varepsilon\right)\right] = \mathbb{E}\left[\mathcal{R}_{\lambda_\varepsilon}(\hat{\Gamma}_\varepsilon) - \mathcal{R}_{\lambda_\varepsilon}(\tilde{\Gamma}_\varepsilon)\right] + \mathbb{E}\left[\mathcal{E}_{\lambda_\varepsilon}\left(\tilde{\Gamma}_\varepsilon\right)\right] , \tag{9}$$

and show that both terms in the r.h.s. go to zero.

• **Step 1.** $\mathbb{E}\left[\mathcal{E}_{\lambda_\varepsilon}\left(\tilde{\Gamma}_\varepsilon\right)\right] \to 0$. We use Proposition 2.4 and get the following result.

**Proposition B.1.** *Let $\varepsilon \in (0, 1)$. Under Assumptions 2.2 and 3.1, the following holds*

$$\mathbb{E}\left[\mathcal{E}_{\lambda_\varepsilon}\left(\tilde{\Gamma}_\varepsilon\right)\right] \leq \mathbb{E}\left[(\hat{f}(X) - f^*(X))^2\right] + \mathbb{E}\left[|\hat{\sigma}^2(X) - \sigma^2(X)|\right] + C\mathbb{E}\left[|F_{\hat{\sigma}^2}(\lambda_\varepsilon) - F_{\sigma^2}(\lambda_\varepsilon)|\right],$$

*where $C > 0$ is constant which depends on the upper bounds of $\sigma^2$ and $\lambda_\varepsilon$.*

*Proof of Proposition B.1.* Let $\varepsilon \in (0, 1)$. First, we recall our notation $F_{\hat{\sigma}^2}(\cdot) = \mathbb{P}_X\left(\hat{\sigma}^2(X) \leq \cdot|\mathcal{D}_n\right)$ and $\lambda_\varepsilon = F_{\sigma^2}^{-1}(1 - \varepsilon)$. We also introduce $\tilde{\lambda}_\varepsilon = F_{\hat{\sigma}^2}^{-1}(1 - \varepsilon)$ for the pseudo-oracle counterpart of $\lambda_\varepsilon$. A direct application of Proposition 2.4 yields

$$\mathcal{E}\left(\tilde{\Gamma}_\varepsilon\right) \leq \mathbb{E}_X\left[\left(\hat{f}(X) - f^*(X)\right)^2\right] + \mathbb{E}_X\left[|\sigma^2(X) - \lambda_\varepsilon|\mathbb{1}_{\{|\tilde{\Gamma}_\varepsilon(X)|\neq|\Gamma_\varepsilon^*(X)|\}}\right]. \tag{10}$$

We first observe that if $\sigma^2(X) \leq \lambda_\varepsilon$ and $\hat{\sigma}^2(X) \geq \tilde{\lambda}_\varepsilon$, we have either of the two cases

- $\tilde{\lambda}_\varepsilon \geq \lambda_\varepsilon$ and then $|\sigma^2(X) - \lambda_\varepsilon| \leq |\hat{\sigma}^2(X) - \sigma^2(X)|$;

- $\tilde{\lambda}_\varepsilon \leq \lambda_\varepsilon$ and then either $|\sigma^2(X) - \lambda_\varepsilon| \leq |\hat{\sigma}^2(X) - \sigma^2(X)|$ or $\hat{\sigma}^2(X) \in (\tilde{\lambda}_\varepsilon, \lambda_\varepsilon)$.

Similar reasoning holds in the case where $\sigma^2(X) \geq \lambda_\varepsilon$ and $\hat{\sigma}^2(X) \leq \tilde{\lambda}_\varepsilon$. Therefore

$$\mathbb{E}\left[|\sigma^2(X) - \lambda_\varepsilon| \mathbb{1}_{\left\{|\tilde{\Gamma}_\varepsilon(X)| \neq |\Gamma_\varepsilon^*(X)|\right\}} | \mathcal{D}_n\right]$$
$$\leq \mathbb{E}\left[|\sigma^2(X) - \lambda_\varepsilon| \mathbb{1}_{\{|\sigma^2(X) - \lambda_\varepsilon| \leq |\hat{\sigma}^2(X) - \sigma^2(X)|\}} | \mathcal{D}_n\right]$$
$$+ \mathbb{1}_{\left\{\lambda_\varepsilon \leq \tilde{\lambda}_\varepsilon\right\}} \mathbb{E}\left[|\sigma^2(X) - \lambda_\varepsilon| \mathbb{1}_{\left\{\lambda_\varepsilon \leq \hat{\sigma}^2(X) \leq \tilde{\lambda}_\varepsilon\right\}} | \mathcal{D}_n\right]$$
$$+ \mathbb{1}_{\left\{\tilde{\lambda}_\varepsilon \leq \lambda_\varepsilon\right\}} \mathbb{E}\left[|\sigma^2(X) - \lambda_\varepsilon| \mathbb{1}_{\left\{\tilde{\lambda}_\varepsilon \leq \hat{\sigma}^2(X) \leq \lambda_\varepsilon\right\}} | \mathcal{D}_n\right] \ .$$

From the above inequality, since $\sigma^2$ is bounded, there exists a constant $C > 0$ such that

$$\mathbb{E}\left[|\sigma^2(X) - \lambda_\varepsilon| \mathbb{1}_{\left\{|\tilde{\Gamma}_\varepsilon(X)| \neq |\Gamma_\varepsilon^*(X)|\right\}}\right] \leq \mathbb{E}\left[|\hat{\sigma}^2(X) - \sigma^2(X)|\right] + C\mathbb{E}\left[|F_{\hat{\sigma}^2}(\tilde{\lambda}_\varepsilon) - F_{\hat{\sigma}^2}(\lambda_\varepsilon)|\right] \ .$$

Now, from Assumptions 2.2 and 3.1, we have that $F_{\sigma^2}(\lambda_\varepsilon) = 1 - \varepsilon = F_{\hat{\sigma}^2}(\tilde{\lambda}_\varepsilon)$. Therefore, we deduce that

$$\mathbb{E}\left[|\sigma^2(X) - \lambda_\varepsilon| \mathbb{1}_{\left\{|\tilde{\Gamma}_\varepsilon(X)| \neq |\Gamma_\varepsilon^*(X)|\right\}}\right] \leq \mathbb{E}\left[|\hat{\sigma}^2(X) - \sigma^2(X)|\right] + C\mathbb{E}\left[|F_{\sigma^2}(\lambda_\varepsilon) - F_{\hat{\sigma}^2}(\lambda_\varepsilon)|\right] \ .$$

Putting this into Equation (10) gives the result in Proposition B.1. $\qquad\square$

Since $\hat{f}$ and $\hat{\sigma}^2$ are consistent *w.r.t.* the $L_2$ and $L_1$ risks respectively, the first two terms in the bound of Proposition B.1 converge to zero. It remains to study the convergence of the last term. To this end, we prove that

$$\mathbb{E}\left[|F_{\sigma^2}(\lambda_\varepsilon) - F_{\hat{\sigma}^2}(\lambda_\varepsilon)|\right] = \mathbb{E}\left[|\mathbb{1}_{\{\sigma^2(X) \leq \lambda_\varepsilon\}} - \mathbb{1}_{\{\hat{\sigma}^2(X) \leq \lambda_\varepsilon\}}|\right]$$
$$\leq \mathbb{P}\left(|\sigma^2(X) - \hat{\sigma}^2(X)| \geq |\sigma^2(X) - \lambda_\varepsilon|\right) \ .$$

Hence, for any $\beta > 0$, using Markov's Inequality we have

$$\mathbb{E}\left[|F_{\sigma^2}(\lambda_\varepsilon) - F_{\hat{\sigma}^2}(\lambda_\varepsilon)|\right] \leq \mathbb{P}\left(|\sigma^2(X) - \lambda_\varepsilon| \leq \beta\right) + \mathbb{P}\left(|\sigma^2(X) - \hat{\sigma}^2(X)| \geq \beta\right)$$
$$\leq \mathbb{P}\left(|\sigma^2(X) - \lambda_\varepsilon| \leq \beta\right) + \frac{\mathbb{E}\left[|\hat{\sigma}^2(X) - \sigma^2(X)|\right]}{\beta} \ .$$

Combining this last inequality with Proposition B.1 and the consistency of $\hat{f}$ and $\hat{\sigma}^2$ *w.r.t.* the $L_2$ and $L_1$ risks respectively implies that for all $\beta > 0$

$$\limsup_{n,N\to+\infty} \mathbb{E}\left[\mathcal{E}_{\lambda_\varepsilon}\left(\tilde{\Gamma}_\varepsilon\right)\right] \leq C\mathbb{P}\left(|\sigma^2(X) - \lambda_\varepsilon| \leq \beta\right) \ .$$

Since the above inequality holds for all $\beta > 0$, under Assumption 2.2, we deduce that

$$\mathbb{E}\left[\mathcal{E}_{\lambda_\varepsilon}\left(\tilde{\Gamma}_\varepsilon\right)\right] \to 0 \ ,$$

and then this step of the proof is complete.

- **Step 2.** $\mathbb{E}\left[\mathcal{R}_{\lambda_\varepsilon}(\hat{\Gamma}_\varepsilon) - \mathcal{R}_{\lambda_\varepsilon}(\tilde{\Gamma}_\varepsilon)\right] \to 0$. Thanks to Equation (6), we have that

$$\mathcal{R}_{\lambda_\varepsilon}(\hat{\Gamma}_\varepsilon) - \mathcal{R}_{\lambda_\varepsilon}(\tilde{\Gamma}_\varepsilon) = \mathbb{E}_X\left[\left\{(f^*(X) - \hat{f}(X))^2 + (\sigma^2(X) - \lambda_\varepsilon)\right\}\left(\mathbb{1}_{\left\{|\hat{\Gamma}_\varepsilon(X)|=1\right\}} - \mathbb{1}_{\left\{|\tilde{\Gamma}_\varepsilon(X)|=1\right\}}\right)\right] \ .$$

Therefore, since $\sigma^2$ is bounded, there exists a constant $C > 0$ such that

$$\mathbb{E}\left[|\mathcal{R}_{\lambda_\varepsilon}(\hat{\Gamma}_\varepsilon) - \mathcal{R}_{\lambda_\varepsilon}(\tilde{\Gamma}_\varepsilon)|\right] \leq 2\mathbb{E}\left[(f^*(X) - \hat{f}(X))^2\right] + C\mathbb{E}\left[|\mathbb{1}_{\left\{|\hat{\Gamma}_\varepsilon(X)|=1\right\}} - \mathbb{1}_{\left\{|\tilde{\Gamma}_\varepsilon(X)|=1\right\}}|\right]$$
$$\leq 2\mathbb{E}\left[(f^*(X) - \hat{f}(X))^2\right] + CA_\varepsilon \ , \tag{11}$$

where

$$A_\varepsilon = \mathbb{E}\left[\left|\mathbb{1}_{\left\{|\hat{\Gamma}_\varepsilon(X)|=1\right\}} - \mathbb{1}_{\left\{|\tilde{\Gamma}_\varepsilon(X)|=1\right\}}\right|\right] = \mathbb{E}\left[\left|\mathbb{1}_{\{\hat{F}_{\hat{\sigma}^2}(\hat{\sigma}^2(X))\geq 1-\varepsilon\}} - \mathbb{1}_{\{F_{\hat{\sigma}^2}(\hat{\sigma}^2(X))\geq 1-\varepsilon\}}\right|\right] \ .$$
(12)

Considering the fact that $\hat{f}$ is consistent *w.r.t.* the $L_2$ risk, it remains to treat the term $A_\varepsilon$. We have

$$A_\varepsilon \leq \mathbb{P}\left(|\hat{F}_{\hat{\sigma}^2}(\hat{\sigma}^2(X)) - F_{\hat{\sigma}^2}(\hat{\sigma}^2(X))| \geq |F_{\hat{\sigma}^2}(\hat{\sigma}^2(X)) - (1-\varepsilon)|\right) \ ,$$

and then, for all $\beta > 0$, the following holds

$$A_\varepsilon \leq \mathbb{P}\left(|F_{\hat{\sigma}^2}(\hat{\sigma}^2(X)) - (1-\varepsilon)| < \beta\right) + \mathbb{P}\left(|\hat{F}_{\hat{\sigma}^2}(\hat{\sigma}^2(X)) - F_{\hat{\sigma}^2}(\hat{\sigma}^2(X))| \geq \beta\right) \ .$$
(13)

Under Assumption 3.1, the random variable $F_{\hat{\sigma}^2}(\hat{\sigma}^2(X))$ is uniformly distributed on $[0, 1]$ conditionally on $\mathcal{D}_n$. Therefore, we deduce that

$$
\begin{aligned}
\mathbb{P}\left(|F_{\hat{\sigma}^2}(\hat{\sigma}^2(X)) - (1-\varepsilon)| < \beta\right) &= \mathbb{E}\left[\mathbb{P}_X\left(|F_{\hat{\sigma}^2}(\hat{\sigma}^2(X)) - (1-\varepsilon)| < \beta\right)|\mathcal{D}_n\right] \\
&= \mathbb{E}\left[2\beta|\mathcal{D}_n\right] = 2\beta \ .
\end{aligned}
$$
(14)

According to the second term in the r.h.s. of Equation (13). we have that

$$
\begin{aligned}
\mathbb{P}\left(|\hat{F}_{\hat{\sigma}^2}(\hat{\sigma}^2(X)) - F_{\hat{\sigma}^2}(\hat{\sigma}^2(X))| \geq \beta\right) &\leq \mathbb{P}\left(\sup_{x\in\mathbb{R}}|\hat{F}_{\hat{\sigma}^2}(x) - F_{\hat{\sigma}^2}(x)| \geq \beta\right) \\
&= \mathbb{E}\left[\mathbb{P}_{\mathcal{D}_N}\left(\sup_{x\in\mathbb{R}}|\hat{F}_{\hat{\sigma}^2}(x) - F_{\hat{\sigma}^2}(x)| \geq \beta|\mathcal{D}_n\right)\right] \ ,
\end{aligned}
$$

where $\mathbb{P}_{\mathcal{D}_N}$ is the probability measure *w.r.t.* the dataset $\mathcal{D}_N$. Since, conditionally on $\mathcal{D}_n$, $\hat{F}_{\hat{\sigma}^2}$ is the empirical counterpart of the continuous cumulative distribution function $F_{\hat{\sigma}^2}$, applying the Dvoretzky-Kiefer-Wolfowitz Inequality [17], we deduce that

$$\mathbb{P}\left(|\hat{F}_{\hat{\sigma}^2}(\hat{\sigma}^2(X)) - F_{\hat{\sigma}^2}(\hat{\sigma}^2(X))| \geq \beta\right) \leq 2\exp(-2N\beta^2) \ .$$
(15)

Putting (14) and (15) into Eq. (13), we have that for all $\beta > 0$

$$A_\varepsilon \leq 2\left(\beta + \exp\left(-2N\beta^2\right)\right) \ .$$
(16)

Since Equation (16) holds for all $\beta > 0$, we have that $A_\varepsilon \to 0$ as $N, n \to +\infty$. Hence, from the above inequality we get the desired result in **Step 2**:

$$\mathbb{E}\left[\left|\mathcal{R}_{\lambda_\varepsilon}(\hat{\Gamma}_\varepsilon) - \mathcal{R}_{\lambda_\varepsilon}(\tilde{\Gamma}_\varepsilon)\right|\right] \to 0 \ .$$

Combining **Step 1** and **Step 2** yields the convergence: $\mathbb{E}\left[\mathcal{E}_{\lambda_\varepsilon}\left(\hat{\Gamma}_\varepsilon\right)\right] \to 0$ as $N, n \to +\infty$.

• **Bound on** $\mathbb{E}\left[r(\hat{\Gamma}_\varepsilon)\right]$. To finish the proof of Theorem 3.2, it remains to control the rejection rate $\mathbb{E}\left[r(\hat{\Gamma}_\varepsilon)\right]$ and show that it satisfies $\mathbb{E}\left[\left|r(\hat{\Gamma}_\varepsilon) - \varepsilon\right|\right] \leq CN^{-1/2}$ for some constant $C > 0$. We observe that

$$\mathbb{E}\left[\left|r(\hat{\Gamma}_\varepsilon) - \varepsilon\right|\right] = \mathbb{E}\left[\left|r(\hat{\Gamma}_\varepsilon) - r(\tilde{\Gamma}_\varepsilon)\right|\right] \leq A_\varepsilon \ ,$$

where $A_\varepsilon$ is given by Eq. (12). Repeating the same reasoning as in **Step 2** above, we bound $A_\varepsilon$ as in Eq. (13), and get from Dvoretsky-Kiefer-Wolfowitz Inequality (see Equation (15)), that for all $\beta > 0$,

$$\mathbb{P}\left(|\hat{F}_{\hat{\sigma}^2}(\hat{\sigma}^2(X)) - F_{\hat{\sigma}^2}(\hat{\sigma}^2(X))| \geq \beta\right) \leq 2\exp(-2N\beta^2) \ ,$$

and from Equation (14),

$$\mathbb{P}\left(|F_{\hat{\sigma}^2}(\hat{\sigma}^2(X)) - (1-\varepsilon)| < \beta\right) = 2\beta \ .$$

These two bounds combined the classical peeling argument of [1] (see Lemma E.1 below) imply the desired result:

$$A_\varepsilon \leq CN^{-1/2} \ .$$
(17)

## C Proof of rates of convergence: Theorem 4.4

In this section, we follow the same strategy as in Section B but here we care about rates of convergence. Moreover, we have to pay attention to the randomness we introduced in the predictor because of the use of $k$NN. As in Section B, we introduce some pseudo-oracle predictor. However, this one needs to depend on the randomness we introduced in the definition of $\hat{\Gamma}_\varepsilon(x, \zeta)$. Define the pseudo-oracle $\varepsilon$-predictor $\tilde{\Gamma}_\varepsilon$ for all $x \in \mathbb{R}^d$ and $\zeta \in [0, u]$ as[3]

$$\tilde{\Gamma}_\varepsilon(x, \zeta) = \begin{cases} \left\{ \hat{f}(x) \right\} & \text{if } \bar{\sigma}^2(x, \zeta) \leq F_{\bar{\sigma}^2}^{-1}(1 - \varepsilon) \\ \emptyset & \text{otherwise .} \end{cases}$$

To study the excess risk $\mathbb{E}\left[ \mathcal{E}_{\lambda_\varepsilon}\left( \hat{\Gamma}_\varepsilon \right) \right]$ of our predictor, we also consider a similar decomposition as in Eq. (9) and treat each of the two terms separately.

● **Step 1.** Study of $\mathbb{E}\left[ \mathcal{E}_{\lambda_\varepsilon}\left( \tilde{\Gamma}_\varepsilon \right) \right]$. We establish the following result.

**Proposition C.1.** *Assume that Assumptions 4.2 and 4.3 are fulfilled for some $\alpha \geq 0$, then the following inequality holds*

$$\mathbb{E}\left[ \mathcal{E}_{\lambda_\varepsilon}\left( \tilde{\Gamma}_\varepsilon \right) \right] \leq \mathbb{E}\left[ \left( f^*(X) - \hat{f}(X) \right)^2 \right] + C \left( \mathbb{E}\left[ \left( \sup_{x \in \mathcal{C}} \left| \hat{\sigma}^2(x) - \sigma^2(x) \right| \right)^{1+\alpha} \right] + u^{1+\alpha} \right) ,$$

*where $C > 0$ depends only on $c^*$ and $\alpha$.*

*Proof.* Let $\varepsilon \in (0, 1)$. We recall that $\lambda_\varepsilon = F_{\sigma^2}^{-1}(1 - \varepsilon)$ and $\tilde{\lambda}_\varepsilon = F_{\bar{\sigma}^2}^{-1}(1 - \varepsilon)$. Since $\zeta$ is distributed according to a Uniform distribution on $[0, u]$, we observe that

$$\left| \bar{\sigma}^2(X, \zeta) - \sigma^2(X) \right| \leq \sup_{x \in \mathcal{C}} \left| \sigma^2(x) - \hat{\sigma}^2(x) \right| + u := \hat{h}_u .$$

Hence, according to Theorem 2.12 in [4] (recalled in Lemma E.2), we have that conditionally on $\mathcal{D}_n$

$$\left| \tilde{\lambda}_\varepsilon - \lambda_\varepsilon \right| \leq \hat{h}_u .$$

Furthermore, since $X$ and $\zeta$ are independent, we can use Proposition 2.4 and get

$$\mathbb{E}\left[ \mathcal{E}_{\lambda_\varepsilon}\left( \tilde{\Gamma}_\varepsilon \right) \right] \leq \mathbb{E}\left[ \left( \hat{f}(X) - f^*(X) \right)^2 \right] + \mathbb{E}\left[ |\sigma^2(X) - \lambda_\varepsilon| \mathbb{1}_{\left\{ |\tilde{\Gamma}_\varepsilon(X, \zeta)| \neq |\Gamma_\varepsilon^*(X)| \right\}} \right] .$$

On the event $\left\{ |\tilde{\Gamma}_\varepsilon(X, \zeta)| \neq |\Gamma_\varepsilon^*(X)| \right\}$, we note that

$$|\sigma^2(X) - \lambda_\varepsilon| \leq |\bar{\sigma}^2(X, \zeta) - \sigma^2(X)| + |\tilde{\lambda}_\varepsilon - \lambda_\varepsilon| .$$

Therefore, conditional on $\mathcal{D}_n$, we deduce the following

$$\begin{aligned} \mathbb{E}_{(X, \zeta)}\left[ |\sigma^2(X) - \lambda_\varepsilon| \mathbb{1}_{\left\{ |\tilde{\Gamma}_\varepsilon(X, \zeta)| \neq |\Gamma_\varepsilon^*(X)| \right\}} \right] &\leq \mathbb{E}_{(X, \zeta)}\left[ |\sigma^2(X) - \lambda_\varepsilon| \mathbb{1}_{\left\{ |\sigma^2(X) - \lambda_\varepsilon| \leq |\bar{\sigma}^2(X, \zeta) - \sigma^2(X)| + |\tilde{\lambda}_\varepsilon - \lambda_\varepsilon| \right\}} \right] \\ &\leq \mathbb{E}_X\left[ |\sigma^2(X) - \lambda_\varepsilon| \mathbb{1}_{\left\{ |\sigma^2(X) - \lambda_\varepsilon| \leq 2\hat{h}_u \right\}} \right] \\ &\leq 2\hat{h}_u \mathbb{P}_X\left( |\sigma^2(X) - \lambda_\varepsilon| \leq 2\hat{h}_u \right) . \end{aligned}$$

Finally, applying Assumption 4.3, we deduce that there exists a constant $C > 0$ such that

$$\mathbb{E}\left[ |\sigma^2(X) - \lambda_\varepsilon| \mathbb{1}_{\left\{ |\tilde{\Gamma}_\varepsilon(X, \zeta)| \neq |\Gamma_\varepsilon^*(X)| \right\}} \right] \leq C \left( \mathbb{E}\left[ \sup_{x \in \mathcal{C}} \left| \sigma^2(x) - \hat{\sigma}^2(x) \right|^{1+\alpha} \right] + u^{1+\alpha} \right) ,$$

which ends the proof. □

Based on Proposition C.1, the control of $\mathbb{E}\left[\mathcal{E}_{\lambda_\varepsilon}\left(\tilde{\Gamma}_\varepsilon\right)\right]$ requires a bound on $\mathbb{E}\left[\left(f^*(X) - \hat{f}(X)\right)^2\right]$ and on $\mathbb{E}\left[\sup_{x\in\mathcal{C}}\left|\sigma^2(x) - \hat{\sigma}^2(x)\right|^{1+\alpha}\right]$. The first of these two terms relies on estimation of the regression function with $k$NN algorithm and is rather well studied. In particular, thanks to Proposition D.4 we have with the choice $k_n \propto n^{2/(d+2)}$

$$\mathbb{E}\left[\left(\hat{f}(X) - f^*(X)\right)^2\right] \leq Cn^{-2/(d+2)} \ , \tag{18}$$

where $C > 0$ is a constant which depends on $f^*$, $c_0$, $\mathcal{C}$, and $d$. Then it remains to bound the second term which is the purpose of Proposition D.5 that relies on the rate of convergence of the $k$NN estimator of the conditional variance $\hat{\sigma}^2$ in supremum norm. This result says that under our assumptions and for the choice $k_n \propto n^{2/(d+2)}$, we have that

$$\mathbb{E}\left[\left(\sup_{x\in\mathcal{C}}\left|\hat{\sigma}^2(x) - \sigma^2(x)\right|\right)^{1+\alpha}\right] \leq C\log(n)^{(\alpha+1)}n^{-(\alpha+1)/(d+2)} \ ,$$

for a constant $C > 0$ that depends on $f^*$, $\sigma^2$, $c_0$, $\mathcal{C}$, and on the dimension $d$. Putting this last inequality and Eq. (18) into the upper bound on the excess risk of $\tilde{\Gamma}_\varepsilon$ from Proposition C.1 we show that when we set $u = u_n \leq n^{-1/(d+2)}$ we can write

$$\mathbb{E}\left[\mathcal{E}_{\lambda_\varepsilon}\left(\tilde{\Gamma}_\varepsilon\right)\right] \leq C\left(n^{-2/(d+2)} + \log(n)^{(\alpha+1)}n^{-(\alpha+1)/(d+2)}\right) \ ,$$

where $C > 0$ is a constant which depends on $f^*$, $\sigma^2$, $c_0$, $c^*$, $\alpha$, $\mathcal{C}$, and on the dimension $d$. This ends the first step of the proof.

• **Step 2.** Study of $\mathbb{E}\left[\mathcal{R}_{\lambda_\varepsilon}(\hat{\Gamma}_\varepsilon) - \mathcal{R}_{\lambda_\varepsilon}(\tilde{\Gamma}_\varepsilon)\right]$. Since $X$ and $\zeta$ are independent, as in **Step 2** of the proof of Theorem 3.2 (*cf.* Eq. (11)), we get

$$\mathbb{E}\left[\left|\mathcal{R}_{\lambda_\varepsilon}(\hat{\Gamma}_\varepsilon) - \mathcal{R}_{\lambda_\varepsilon}(\tilde{\Gamma}_\varepsilon)\right|\right] \leq 2\mathbb{E}\left[\left(f^*(X) - \hat{f}(X)\right)^2\right] + CA_\varepsilon \ ,$$

where $A_\varepsilon$ is defined similarly as in Equation (12) with a small modification due to the random perturbation we made on $\hat{\sigma}^2$. Similarly we have

$$A_\varepsilon \leq \mathbb{P}\left(\left|\hat{F}_{\bar{\sigma}^2}(\bar{\sigma}^2(X,\zeta)) - \{F_{\bar{\sigma}^2}(\bar{\sigma}^2(X,\zeta))\right| \geq \left|F_{\bar{\sigma}^2}(\bar{\sigma}^2(X,\zeta)) - (1-\varepsilon)\right|\right) \ .$$

Therefore using the same arguments as in **Step 2** of the proof of Theorem 3.2 to get (17), it is easy to see that there exists $C > 0$ such that $A_\varepsilon \leq CN^{-1/2}$. Then, we deduce

$$\mathbb{E}\left[\left|\mathcal{R}_{\lambda_\varepsilon}(\hat{\Gamma}_\varepsilon) - \mathcal{R}_{\lambda_\varepsilon}(\tilde{\Gamma}_\varepsilon)\right|\right] \leq 2\mathbb{E}\left[\left(f^*(X) - \hat{f}(X)\right)^2\right] + CN^{-1/2}.$$

Finally, an application of Theorem D.4 yields

$$\mathbb{E}\left[\left|\mathcal{R}_{\lambda_\varepsilon}(\hat{\Gamma}_\varepsilon) - \mathcal{R}_{\lambda_\varepsilon}(\tilde{\Gamma}_\varepsilon)\right|\right] \leq C\left(n^{-2/(d+2)} + N^{-1/2}\right),$$

where $C > 0$ is a constant which depends on $f^*$, $c_0$, $\mathcal{C}$, and $d$. This ends **Step 2** of the proof.

Lastly, we combine the results in **Step 1** and **Step 2**, together with the decomposition

$$\mathbb{E}\left[\mathcal{E}_{\lambda_\varepsilon}\left(\hat{\Gamma}_\varepsilon\right)\right] = \mathbb{E}\left[\mathcal{R}_{\lambda_\varepsilon}(\hat{\Gamma}_\varepsilon) - \mathcal{R}_{\lambda_\varepsilon}(\tilde{\Gamma}_\varepsilon)\right] + \mathbb{E}\left[\mathcal{E}_{\lambda_\varepsilon}\left(\tilde{\Gamma}_\varepsilon\right)\right] \ ,$$

and get the desired bound on the excess risk.

## D Rate of convergence for $k$NN estimator

In this section , we focus on rates of convergence of $k$NN for the estimation of the regression function $f^*$ and the conditional variance function $\sigma^2$. The proofs techniques are largely inspired by those in [3, 10], though we provide some additional steps to build for instance finite sample bounds for the sup norm in the problem of conditional variance estimation.

## D.1 Regression function estimation

We provide the rate of convergence of the $k$NN estimator of $f^*$ in the regression model for which we make the following assumptions. We assume that $f^*$ is Lipschitz (Assumption 4.1) and that Assumption 4.2 are fulfilled. We recall that from Assumption 4.2, we have that $\mathbb{P}_X$ is supported on a compact set $\mathcal{C}$. Furthermore, we also assume that $Y - f^*(X)$ satisfies a uniform noise condition: there exists $c_0 > 0$ such that

$$\sup_{x \in \mathcal{C}} \mathbb{E}\left[\exp(\lambda\,(Y - f^*(X)))\mid X = x\right] \le \exp(c_0^2 \lambda^2), \quad \text{for } |\lambda| \le \frac{1}{c_0} \; . \tag{19}$$

This assumption is rather weak and requires that conditional on $X$ is sub-exponential uniformly over $\mathcal{C}$ (see [24]). Using the same notation as in Section 3, we recall that the $k$NN estimator $\hat{f}$ of $f$ is defined as follows

$$\hat{f}(x) = \frac{1}{k_n} \sum_{i=1}^{k_n} Y_{(i,n)}(x) \; .$$

The purpose of the appendix is to provide rates of convergence for the $k$NN estimator $\hat{f}$ under the above assumption. To this end we require two auxiliary lemmata, which provide a control respectively with high probability and in expectation on the distance between a feature point and its neighbors uniformly over $\mathcal{C}$.

**Lemma D.1.** *Assume Assumptions 4.1-4.2 hold. Then there exist $C_1 > 0$, which depends only on $\mathcal{C}, \mu_{\min}$, and on $d$ and $C_2 > 0$, which depends on $\mathcal{C}$ and on $d$, such that for all $t \ge \left(\frac{\log(n) k_n}{C_1 n}\right)^{1/d}$, we have*

$$\mathbb{P}\left(\sup_{x \in \mathcal{C}} \frac{1}{k_n} \sum_{i=1}^{k_n} \left\|X_{(i,n)}(x) - x\right\| \ge t\right) \le C_2 \exp\left(\log(n) - C_1 t^d n / k_n\right) \; .$$

*Proof.* For any $a \in \mathbb{R}$, let us denote by $\lfloor a \rfloor$ the largest integer which is smaller or equal to $a$. Consider some $x \in \mathcal{C}$. Following the same arguments as in proof of Theorem 6.2 in [10], we split the data $X_1, \ldots, X_n$ into $k_n + 1$ folds such that the first $k_n$ folds have the same size $\lfloor \frac{n}{k_n} \rfloor$ and the last fold contains the remaining data if there are. We denote $\tilde{X}_j^x$ the nearest neighbor of $x$ in the $j$th fold and then obviously

$$\sum_{i=1}^{k_n} \left\|X_{(i,n)}(x) - x\right\| \le \sum_{j=1}^{k_n} \left\|\tilde{X}_j^x - x\right\| \; .$$

Let $\bar{B}_2(a, r)$ be the closed Euclidean ball in $\mathbb{R}^d$ centered in $a$ with radius $r > 0$. Since $\mathcal{C}$ is compact, we have $\mathcal{C} \subset \bar{B}_2(0, R)$ for some $R > 0$, and therefore there exists an $\varepsilon$-net $\mathcal{C}_\varepsilon$ of $\mathcal{C}$ w.r.t. $\|.\|$ such that $|\mathcal{C}_\varepsilon| \le \left(\frac{3R}{\varepsilon}\right)^d$. In particular, for all $x \in \mathcal{C}$ there exists $x_\varepsilon \in \mathcal{C}_\varepsilon$ such that $\|x - x_\varepsilon\| \le \varepsilon$. Then, for all $x \in \mathcal{C}$ and all $j \in \{1, \ldots, k_n\}$, there exists $x_\varepsilon \in \mathcal{C}_\varepsilon$ such that

$$\left\|\tilde{X}_j^x - x\right\| \le \left\|\tilde{X}_j^x - x_\varepsilon\right\| + \varepsilon \; . \tag{20}$$

Besides, we observe that

$$\left\|\tilde{X}_j^x - x_\varepsilon\right\| \le \left\|\tilde{X}_j^{x_\varepsilon} - x_\varepsilon\right\| + 2\varepsilon \; . \tag{21}$$

Indeed, if $\left\|\tilde{X}_j^{x_\varepsilon} - x_\varepsilon\right\| + 2\varepsilon < \left\|\tilde{X}_j^x - x_\varepsilon\right\|$ we can write

$$\begin{aligned}
\left\|\tilde{X}_j^{x_\varepsilon} - x\right\| + \varepsilon &\le \left\|\tilde{X}_j^{x_\varepsilon} - x_\varepsilon\right\| + 2\varepsilon \\
&< \left\|\tilde{X}_j^x - x_\varepsilon\right\| \le \left\|\tilde{X}_j^x - x\right\| + \varepsilon \; ,
\end{aligned}$$

which contradicts the fact that $\tilde{X}_j^x$ is the nearest neighbor of $x$ in the $j$th fold. Hence, from Equations (20) and (21), we deduce that

$$\sup_{x \in \mathcal{C}} \frac{1}{k_n} \sum_{i=1}^{k_n} \left\|X_{(i,n)}(x) - x\right\| \le 3\varepsilon + \sup_{x \in \mathcal{C}_\varepsilon} \frac{1}{k_n} \sum_{j=1}^{k_n} \left\|\tilde{X}_j^x - x\right\| \; .$$

From the above inequality, we obtain that for $t > 6\varepsilon$,

$$\mathbb{P}\left(\sup_{x\in\mathcal{C}}\frac{1}{k_n}\sum_{i=1}^{k_n}\left\|X_{(i,n)}(x)-x\right\|\geq t\right)\leq\mathbb{P}\left(\sup_{x\in\mathcal{C}_\varepsilon}\frac{1}{k_n}\sum_{j=1}^{k_n}\left\|\tilde{X}_j^x-x\right\|\geq t/2\right) \quad . \tag{22}$$

Our goal becomes to bound r.h.s. of the above inequality. Using union bound, we deduce that for all $t > 6\varepsilon$

$$\mathbb{P}\left(\sup_{x\in\mathcal{C}_\varepsilon}\frac{1}{k_n}\sum_{j=1}^{k_n}\left\|\tilde{X}_j^x-x\right\|\geq t/2\right)\leq\sum_{x\in\mathcal{C}_\varepsilon}\mathbb{P}\left(\frac{1}{k_n}\sum_{j=1}^{k_n}\left\|\tilde{X}_j^x-x\right\|\geq t/2\right)$$

$$\leq\sum_{x\in\mathcal{C}_\varepsilon}\sum_{j=1}^{k_n}\mathbb{P}\left(\left\|\tilde{X}_j^x-x\right\|\geq t/2\right) \quad . \tag{23}$$

For each $x\in\mathcal{C}_\varepsilon$ and $j\in\{1,\ldots,k_n\}$, by definition of $\tilde{X}_j^x$ and since $(X_i)_{i=1,\ldots,n}$ are i.i.d., we have

$$\mathbb{P}\left(\left\|\tilde{X}_j^x-x\right\|\geq t/2\right)=\mathbb{P}\left(\left\|X_{(1,\lfloor\frac{n}{k_n}\rfloor)}-x\right\|\geq t/2\right)=\left(\mathbb{P}\left(\|X_1-x\|\geq t/2\right)\right)^{\lfloor n/k_n\rfloor} \quad . \tag{24}$$

On one hand, observe that for $t\geq 4R$, $(\mathbb{P}\left(\|X_1-x\|\geq t/2\right)=0$. On the other hand for $t\leq 4R$, using the elementary inequality $\log(1-a)\leq -a$ for all $a\in[0,1)$, we have that

$$\left(\mathbb{P}\left(\|X_1-x\|\geq t/2\right)\right)^{\lfloor n/k_n\rfloor}\leq\exp\left(-\left\lfloor\frac{n}{k_n}\right\rfloor\mathbb{P}\left(\|X_1-x\|\leq t/2\right)\right) \quad ,$$

which yields, thanks to Assumption 4.2, there exists $C>0$ which depends on $\mu_{\min}$ and $d$ such that

$$\left(\mathbb{P}\left(\|X_1-x\|\geq t/2\right)\right)^{\lfloor n/k_n\rfloor}\leq\exp\left(-Ct^d\lfloor n/k_n\rfloor\right) \quad .$$

We finally deduce from Equation (22), (23), and (24), that for all $t\geq 6\varepsilon$

$$\mathbb{P}\left(\sup_{x\in\mathcal{C}}\frac{1}{k_n}\sum_{i=1}^{k_n}\left\|X_{(i,n)}(x)-x\right\|\geq t\right)\leq k_n|\mathcal{C}_\varepsilon|\exp\left(-Ct^d\lfloor n/k_n\rfloor\right) \quad .$$

Choosing $\varepsilon=(\frac{k_n}{6^d Cn})^{1/d}$, we get that for $t\geq(\frac{k_n}{Cn})^{1/d}$,

$$\mathbb{P}\left(\sup_{x\in\mathcal{C}}\frac{1}{k_n}\sum_{i=1}^{k_n}\left\|X_{(i,n)}(x)-x\right\|\geq t\right)\leq C_2\exp\left(\log(n)-C_1 t^d n/k_n\right),$$

which yields the expected result. $\qquad\square$

The second lemma establishes a control in expectation of the uniform distance.

**Lemma D.2.** *Under Assumption 4.2, there exist $C>0$, which depends only on $\mathcal{C},\mu_{\min}$, and on $d$ such that*

$$\mathbb{E}\left[\left(\sup_{x\in\mathcal{C}}\frac{1}{k_n}\sum_{i=1}^{k_n}\left\|X_{(i,n)}(x)-x\right\|\right)^p\right]\leq C\left(\frac{k_n\log(n)}{n}\right)^{p/d} \quad .$$

*Proof.* Since Assumption 4.2 holds, we can use Lemma D.1. Then there exist two non negative constants $C_1$ and $C_2$ such that for all $t\geq\left(\frac{\log(n)k_n}{C_1 n}\right)^{1/d}$, we have

$$\mathbb{P}\left(\sup_{x\in\mathcal{C}}\frac{1}{k_n}\sum_{i=1}^{k_n}\left\|X_{(i,n)}(x)-x\right\|\geq t\right)\leq C_2\exp\left(\log(n)-C_1 t^d n/k_n\right) \quad .$$

Therefore an application of Lemma E.3 implies directly the result. $\qquad\square$

Below, we state the main result of this section related to the rate of convergence in sup norm of the $k$NN estimator of the regression function.

**Theorem D.3.** *Assume Assumption 4.2 is satisfied. Moreover, let $p \geq 1$ and $k_n \propto n^{2/(d+2)}$. Then*

$$\mathbb{E}\left[\left(\sup_{x \in \mathcal{C}}\left|\hat{f}(x) - f^*(x)\right|\right)^p\right] \leq C\log(n)^p n^{-p/(d+2)},$$

*where $C > 0$ is a constant which depends on $f^*$, $c_0$, $\mathcal{C}$, $\mu_{\min}$ and d.*

*Proof.* First, we have that

$$\hat{f}(x) - f^*(x) = \frac{1}{k_n}\sum_{i=1}^{k_n}\left(Y_{(i,n)}(x) - f^*(X_{(i,n)}(x))\right) + \frac{1}{k_n}\sum_{i=1}^{k_n}\left(f^*(X_{(i,n)}(x)) - f^*(x)\right) \quad .$$

Therefore, since $f^*$ is $L$-Lipschitz, we then deduce that

$$\sup_{x \in \mathcal{C}}\left|\hat{f}(x) - f^*(x)\right| \leq \sup_{x \in \mathcal{C}}\left|\frac{1}{k_n}\sum_{i=1}^{k_n}Y_{(i,n)}(x) - f^*(X_{(i,n)}(x))\right| + L\sup_{x \in \mathcal{C}}\frac{1}{k_n}\sum_{i=1}^{k_n}\left\|X_{(i,n)}(x) - x\right\| \quad ,$$

which implies that

$$\mathbb{E}\left[\left(\sup_{x \in \mathcal{C}}\left|\hat{f}(x) - f^*(x)\right|\right)^p\right] \leq 2^{p-1}\mathbb{E}\left[\left(\sup_{x \in \mathcal{C}}\left|\frac{1}{k_n}\sum_{i=1}^{k_n}Y_{(i,n)}(x) - f^*(X_{(i,n)}(x))\right|\right)^p\right]$$

$$+ 2^{p-1}L^p\mathbb{E}\left[\left(\sup_{x \in \mathcal{C}}\frac{1}{k_n}\sum_{i=1}^{k_n}\left\|X_{(i,n)}(x) - x\right\|\right)^p\right] \quad . \quad (25)$$

Lemma D.2 provides a bound on the second term in the r.h.s. of the above inequality. Then it remains to study the first term in the r.h.s. of Eq. (25). Let $x \in \mathcal{C}$, and denote by $\mathcal{N}_{k_n}(x) = \{X_{(1,n)}(x), \ldots X_{(k_n,n)}(x)\}$ the set of the $k_n$-nearest neighbors of $x$ among $\{X_1, \ldots, X_n\}$. We denote by $\mathcal{B}$ the set of all closed balls in $\mathbb{R}^d$. We observe that there exists $\rho_x > 0$ such that $\mathcal{N}_{k_n}(x) \subset \{\bar{B}(x, \rho_x) \cap \{X_1, \ldots, X_n\}\}$, where $\bar{B}(x, \rho_x)$ is the closed ball centered on $x$ with radius $\rho_x$. Therefore

$$\{\mathcal{N}_{k_n}(x), \ x \in \mathcal{C}\} \subset \{\{X_1, \ldots, X_n\} \cap B, \ B \in \mathcal{B}\} \quad .$$

Besides, since the VC-dimension of the class of balls in $\mathbb{R}^d$ is upper bounded by $d + 2$ (see for instance Corollary 13.2 in [8]), Sauer Lemma implies that

$$|\{\{X_1, \ldots, X_n\} \cap B, \ B \in \mathcal{B}\}| \leq \mathcal{S}(\mathcal{B}, n) \leq (n+1)^{d+2} \quad ,$$

where $\mathcal{S}(\mathcal{B}, n)$ denotes the shatter coefficient of $\mathcal{B}$ by $n$ points from $\mathcal{C}$. We then deduce that $|\{\mathcal{N}_{k_n}(x), \ x \in \mathcal{C}\}| \leq (n+1)^{d+2}$, which implies in turn that there exists $\{x_1, \ldots, x_J\}$, with $J \leq (n+1)^{d+2}$ such that

$$\mathbb{E}\left[\left(\sup_{x \in \mathcal{C}}\left|\frac{1}{k_n}\sum_{i=1}^{k_n}Y_{(i,n)}(x) - f^*(X_{(i,n)}(x))\right|\right)^p\right]$$

$$\leq \mathbb{E}\left[\left(\max_{j \in \{1,\ldots,J\}}\left|\frac{1}{k_n}\sum_{i=1}^{k_n}Y_{(i,n)}(x_j) - f^*(X_{(i,n)}(x_j))\right|\right)^p\right].$$

Notice that conditional on $X_1, \ldots, X_n$ the random variables $(Y_{(i,n)}(x_j) - f^*(X_{(i,n)}(x_j))_{i=1,\ldots,k_n}$ are independent with zero mean (see Proposition 8.1 in [3]). Besides from Equation (19) they are uniformly sub-exponential over $\mathcal{C}$, then we deduce from the Bernstein Inequality (see [24]) that for all $t \geq 0$ and $j = 1, \ldots, J$,

$$\mathbb{P}\left(\left|\frac{1}{k_n}\sum_{i=1}^{k_n}Y_{(i,n)}(x_j) - f^*(X_{(i,n)}(x_j))\right| \geq t\right) \leq \exp\left(-ck_n\min\left(\frac{t^2}{K^2}, \frac{t}{K}\right)\right) \quad ,$$

where $c > 0$ is an absolute constant and $K > 0$ depends on $c_0$ in Eq. (19). Set $v_n = \sqrt{\frac{(d+2)\log(n+1)}{ck_n}}$. Our choice of $k_n$ ensures that $v_n \leq 1$, and then we deduce from the union bound that for $t \in (Kv_n, K)$,

$$\mathbb{P}\left(\max_{j \in \{1,\ldots,J\}}\left|\frac{1}{k_n}\sum_{i=1}^{k_n}Y_{(i,n)}(x_j) - f^*(X_{(i,n)}(x_j))\right| \geq t\right) \leq \exp\left((d+2)\log(n+1) - ck_nt^2/K^2\right) \quad ,$$

and for $t > K$,

$$\mathbb{P}\left(\max_{j\in\{1,\ldots,J\}}\left|\frac{1}{k_n}\sum_{i=1}^{k_n}Y_{(i,n)}(x_j)-f^*(X_{(i,n)}(x_j))\right|\geq t\right)\leq\exp\left((d+2)\log(n+1)-ck_nt/K\right) .$$

Considering these two cases, we can derive an exponential bound on the term $\mathbb{P}\left(\max_{j\in\{1,\ldots,J\}}\left|\frac{1}{k_n}\sum_{i=1}^{k_n}Y_{(i,n)}(x_j)-f^*(X_{(i,n)}(x_j))\right|\geq t\right)$ for all $t \geq Kv_n$, therefore we can use similar arguments as in Lemma E.3 and conclude that

$$\mathbb{E}\left[\left(\sup_{x\in\mathcal{C}}\left|\frac{1}{k_n}\sum_{i=1}^{k_n}Y_{(i,n)}(x)-f^*(X_{(i,n)}(x))\right|\right)^p\right]$$

$$\leq\mathbb{E}\left[\left(\max_{j\in\{1,\ldots,J\}}\left|\frac{1}{k_n}\sum_{i=1}^{k_n}Y_{(i,n)}(x_j)-f^*(X_{(i,n)}(x_j))\right|\right)^p\right]\leq C\left(\frac{\log(n)}{k_n}\right)^{p/2} . \quad (26)$$

Combining the above inequality, Equation (25), and Lemma D.2, gives the desired result. $\qquad\square$

To conclude this section, we also provide the rate of convergence of the $k$NN estimator in $L_2$-norm

**Theorem D.4.** *Assume Assumption 4.2 is satisfied and let $k_n \propto n^{2/(d+2)}$, then*

$$\mathbb{E}\left[\left(\hat{f}(X)-f^*(X)\right)^2\right]\leq Cn^{-2/(d+2)} ,$$

*where $C > 0$ is a constant which depends on $f^*$, $c_0$, $\mathcal{C}$, and $d$.*

The proof of this result is provided in [10] for $d \geq 3$ (see Theorem 6.2). However, a small change implies that the same proof holds for all $d$ under Assumption 4.2.

### D.2 Conditional variance function estimation

We provide the rate of convergence of the $k$NN estimator of $\sigma^2$. This proof is largely inspired by [3], though we are interested here in finite sample bounds.

**Proposition D.5.** *Grant Assumptions 4.1 and 4.2. Let $k_n \propto n^{2/(d+2)}$, the following holds*

$$\mathbb{E}\left[\left(\sup_{x\in\mathcal{C}}\left|\hat{\sigma}^2(x)-\sigma^2(x)\right|\right)^{1+\alpha}\right]\leq C\log(n)^{(\alpha+1)}n^{-(\alpha+1)/(d+2)} ,$$

*for all $\alpha \geq 0$, where $C > 0$ is a constant which depends on $f^*$, $\sigma^2$, $c_0$, $\mathcal{C}$, and on the dimension $d$.*

*Proof.* First, we define the function $\tilde{\sigma}^2$ by

$$\tilde{\sigma}^2(x)=\frac{1}{k_n}\sum_{i=1}^{k_n}\left(Y_{(i,n)}(x)-f^*(X_{(i,n)}(x))\right)^2, \quad\forall x\in\mathbb{R}^d .$$

The function $\tilde{\sigma}^2$ is the pseudo-estimator of $\sigma^2$ that would be used in the case where the function $f^*$ is known. By the triangle inequality, we have that for all $x \in \mathcal{C}$,

$$\left|\hat{\sigma}^2(x)-\sigma^2(x)\right|\leq\left|\hat{\sigma}^2(x)-\tilde{\sigma}^2(x)\right|+\left|\tilde{\sigma}^2(x)-\sigma^2(x)\right| .$$

Now, we observe that

$$\hat{\sigma}^2(x)-\tilde{\sigma}^2(x)=$$
$$\frac{1}{k_n}\sum_{i=1}^{k_n}\left(f^*(X_{(i,n)}(x))-\hat{f}(X_{(i,n)}(x))\right)\left(2(Y_{(i,n)}(x)-f^*(X_{(i,n)}(x)))+f^*(X_{(i,n)}(x))-\hat{f}(X_{(i,n)}(x))\right) .$$

Therefore, we deduce

$$\sup_{x\in\mathcal{C}}\left|\hat{\sigma}^2(x)-\sigma^2(x)\right| \le \sup_{x\in\mathcal{C}}\left|\tilde{\sigma}^2(x)-\sigma^2(x)\right| + \left(\sup_{x\in\mathcal{C}}\left|\hat{f}(x)-f^*(x)\right|\right)^2 +$$

$$2\sup_{x\in\mathcal{C}}\left|\hat{f}(x)-f^*(x)\right|\sup_{x\in\mathcal{C}}\left|\frac{1}{k_n}\sum_{i=1}^{k_n}(Y_{(i,n)}(x)-f^*(X_{(i,n)}(x)))\right| \ .$$

From the above inequality, using the fact that $(a+b+c)^p \le 3^{p-1}(a^p+b^p+c^p)$ for $p\ge 1$, $a$, $b$, $c\in\mathbb{R}$ and applying the Cauchy-Schwartz Inequality, we obtain

$$\mathbb{E}\left[\left(\sup_{x\in\mathcal{C}}\left|\hat{\sigma}^2(x)-\sigma^2(x)\right|\right)^{1+\alpha}\right] \le$$

$$C_1\mathbb{E}\left[\left(\sup_{x\in\mathcal{C}}\left|\tilde{\sigma}^2(x)-\sigma^2(x)\right|\right)^{1+\alpha}\right] + C_2\mathbb{E}\left[\left(\sup_{x\in\mathcal{C}}\left|\hat{f}(x)-f^*(x)\right|\right)^{2(1+\alpha)}\right]$$

$$+C_3\left\{\mathbb{E}\left[\left(\sup_{x\in\mathcal{C}}\left|\hat{f}(x)-f^*(x)\right|\right)^{2(1+\alpha)}\right]\right\}^{1/2}\left\{\mathbb{E}\left[\left(\sup_{x\in\mathcal{C}}\left|\frac{1}{k_n}\sum_{i=1}^{k_n}(Y_{(i,n)}(x)-f^*(X_{(i,n)}(x)))\right|\right)^{2(1+\alpha)}\right]\right\}^{1/2} \ ,$$

where $C_1$, $C_2$ and $C_3$ are non negative reals. We finish the proof of the proposition by bounded the above l.h.s. This relies on controls of estimation error of $k$NN for the regression function $f^*$ and the conditional variance function $\sigma^2$. Observe that when $Y$ is either bounded or satisfies the model conditions in Eq. (5), we have that the random variables $Y-f^*(X)$ and $(Y-f^*(X))^2 - \sigma^2(X)$ satisfy the uniform noise condition (19). Indeed, while this fact is clear for $Y-f^*(X)$, it also holds true for $(Y-f^*(X))^2 - \sigma^2(X)$ since, conditionally on $X$, this random variable is either bounded (since $\sigma^2$ is bounded as well) or sub-exponential. Therefore, the result of Theorem D.3 applies for the $k$NN estimators $\tilde{\sigma}^2$ and $\hat{f}$. Furthermore, using the result in Eq. (26), we deduce from the above inequality

$$\mathbb{E}\left[\left(\sup_{x\in\mathcal{C}}\left|\hat{\sigma}^2(x)-\sigma^2(x)\right|\right)^{1+\alpha}\right] \le C\log(n)^{(\alpha+1)}n^{-(\alpha+1)/(d+2)} \ ,$$

where $C>0$ is a constant which depends on $f^*$, $\sigma^2$, $c_0$, $\mathcal{C}$, and the dimension $d$. $\square$

## E  Technical tools

In this section, we state several results that may help for readability of the paper. The first result is a direct application of the classical peeling argument of [1].

**Lemma E.1** (Lemma 1 in [7])**.** *Let $X$ be a real random variable, $(X_n)_{n\ge 1}$ be a sequence of real random variables and $t_0\in\mathbb{R}$. Assume that there exist $C_1>0$ and $\gamma_0>0$ such that*

$$\mathbb{P}_X\left(|X-t_0|\le\delta\right)\le C_1\delta^{\gamma_0}, \quad \forall\delta>0 \ ,$$

*and a sequence of positive numbers $a_n$ tends towards infinity, $C_2$, $C_3$ some positive constants such that*

$$\mathbb{P}_{X_n}\left(|X_n-X|\ge\delta|X\right)\le C_2\exp\left(-C_3 a_n\delta^2\right), \quad \forall\delta>0, \quad \forall n\in\mathbb{N}.$$

*Then, there exists $C>0$ depending only on $C_1, C_2$ and $C_3$, such that*

$$\left|\mathbb{E}\left[\mathbb{1}_{X_n\ge t_0}-\mathbb{1}_{X\ge t_0}\right]\right|\le C a_n^{-\gamma_0/2}.$$

The next result describes the representation of $\infty$-Wasserstein distance ($W_\infty$) on the real line. Let $Z_\infty(\mathbb{R})$ be the collection of all compactly supported probability measures on $\mathbb{R}$.

**Lemma E.2** (Theorem 2.12 in [4])**.** *Let $\mu$ and $\nu$ be probability measures in $Z_\infty(\mathbb{R})$ with respective distribution functions $F$ and $G$. Then, $W_\infty(\mu,\nu) := \sup_{0<t<1}|F^{-1}(t)-G^{-1}(t)|$ is the infimum over all $h\ge 0$ such that*

$$G(x-h)\le F(x)\le G(x+h) \ \text{for all} \ x\in\mathbb{R}.$$

The following result provides a bound on moments of a positive random variable provided a tail control.

**Lemma E.3.** *Let $a \geq 1$, let $b$, $c$ be two non negative real numbers, and let $m \in \mathbb{N}$. Consider $Z$ a positive random variable such that*

$$\mathbb{P}\left(Z \geq t\right) \leq c \exp\left(a - bt^m\right) \ ,$$

*for all $t \geq (a/b)^{1/m}$. Then for all $p \geq 1$, there exists a constant $C > 0$ such that*

$$\mathbb{E}\left[Z^p\right] \leq C(a/b)^{p/m} \ .$$

*Proof.* Using the following equality which holds for any positive random variable $Z$, and any $p \geq 1$

$$\mathbb{E}\left[Z^p\right] = \int_0^{+\infty} \mathbb{P}\left(Z \geq t\right) pt^{p-1}\mathrm{d}t \ , \tag{27}$$

and the condition in Lemma E.3, we deduce

$$\mathbb{E}\left[Z^p\right] \leq \int_0^u pt^{p-1}\mathrm{d}t + c \int_u^{+\infty} \exp\left(a - bt^m\right)pt^{p-1}\mathrm{d}t \ , \tag{28}$$

where $u = (a/b)^{1/m}$ and where we used the trivial inequality $\mathbb{P}\left(Z \geq t\right) \leq 1$ to bound the first term in the r.h.s. Since $(a')^m - (b')^m \geq (a' - b')^m$ for all $a'$, $b' \in \mathbb{R}$ such that $a' \geq b' \geq 0$, we can write that

$$\exp\left(a - bt^m\right) \leq \exp\left(-(t-u)^m b\right) \ ,$$

which yields

$$
\begin{aligned}
\int_u^{+\infty} \exp\left(a - bt^m\right)pt^{p-1}\mathrm{d}t &\leq \int_u^{+\infty} \exp\left(-(t-u)^m b\right) pt^{p-1}\mathrm{d}t \\
&\leq \frac{1}{u} \int_u^{+\infty} \exp\left(-(t-u)^m b\right) pt^p\mathrm{d}t \\
&= \frac{p}{u}\left(\frac{1}{b}\right)^{1/m} \int_0^{+\infty} e^{-v^m} \left(v\left(\frac{1}{b}\right)^{1/m} + u\right)^p \mathrm{d}v \ ,
\end{aligned}
$$

where we consider the changing of variable $v = ((t-u)^m b)^{1/m}$ in the last equality. Finally, using that $(a' + b')^p \leq 2^{p-1}((a')^p + (b')^p)$ for all $p \geq 1$, $a'$, $b' \in \mathbb{R}$ and given that $u \geq (1/b)^{1/m}$, we show from the above inequality that

$$
\begin{aligned}
\int_u^{+\infty} \exp\left(a - bt^m\right)pt^{p-1}\mathrm{d}t &\leq C_1 \left(\frac{1}{b}\right)^{p/m} \int_0^{+\infty} v^p e^{-v^m}\mathrm{d}v + C_2 u^p \int_0^{+\infty} e^{-v^m}\mathrm{d}v \\
&\leq C_3 u^p \ ,
\end{aligned}
$$

for positive constants $C_1$, $C_2$, $C_3$. Inject this into Eq.(28) leads to the result. $\qquad\square$