[Reviews · NeurIPS 2020]

Review 1

Summary and Contributions: This paper consider a regression with reject option problem, where one may abstain from predicting at some "hard" instances, with an emphasis on the case where the rejection (abstention) rate is prescribed. The first contribution is a characterization of the optimal prediction rule (knowing the true distribution of the data) given the rejection rate epsilon, which is obtained by predicting using the regression function, and abstaining when the conditional variance at the input point exceeds its (1-epsilon)-quantile. (This is done by first considering a variant where rejection is associated to a fixed penalty, then using the standard correspondence between penalized and constrained problems.) Motivated by this characterization, the authors propose a plug-in approach, which relies on (1) an estimator of the regression function, (2) an estimator of the conditional variance and (3) an estimator of the quantiles of the conditional variance (taken to be the empirical quantile of the estimated conditional variance on a separate set of data inputs). This plug-in approach is shown to be "consistent" (in that its prediction accuracy and rejection rate converge to that of the best predictor with prescribed rejection rate), provided that the previous estimators are consistent in appropriate senses (L^2 for regression function and L^1 for the conditional variance). Finally, the plug-in approach is applied to the k-Nearest Neighbors (k-NN) algorithm, for which nonparametric rates of convergence for Lipschitz regression function and conditional variance (and some "margin condition" describing the mass of the conditional variance around the optimal threshold) are provided using convergence rates of k-NN. Experiments are provided, showing the performance of the plug-in method for different base regression procedures (SVM, Random forests, k-NN).

Strengths: The problem of regression with reject option is a relevant variation on the problem of prediction with reject option, which has seemingly mostly been considered in the classification case in the literature. This work provides a clean and theoretically grounded treatment of the proposed problem, relying on a basic plug-in approach. [Disclaimer: the problem of prediction with reject option is new to me, and I know little about the related literature.]

Weaknesses: The depth of the contribution is perhaps limited by the relative simplicity of the considered problem, and its relation to the well-studied classification case. For instance, the characterization of the optimal prediction rule in terms of the conditional variance (which is emphasized in the text) is interesting, but quite straightforward, and a direct analogue of the results in the classification case. The consistency results also follow quite directly from the error decomposition.

Correctness: The claims and methods seem correct and are well justified: I checked the proofs in appendices A and B and found no issue, and quickly skimmed over the other proofs which seemed sound as well.

Clarity: The paper is very clear, and the main ideas are nicely conveyed in the text.

Relation to Prior Work: Previous work is acknowledged and discussed, most of it being on the related but distinct classification with rejection problem. (It is likely that all the results presented here have direct analogues for classification, it would be helpful to provide specific pointers to the literature in each case.)

Reproducibility: Yes

Additional Feedback: - It is worth noting that, in the typical case alpha = 1 (namely, the mass of the conditional variance at the considered threshold is not significantly larger or smaller at the desired threshold than at a generic point), the term corresponding to the conditional variance estimation/choice of rejection zone is of the same order (up to an extra logarithmic term) as the term corresponding to the estimation of the regression function. - About the rates of convergence for k-NN (proof in Appendix D): do the results follow from the rates of convergence of k-NN (without reject option) and the previous decomposition (appendix C), or are there specific complications in this case? In the first case, is there no canonical reference providing such rates (which seem to be derived in the proof)? -- Overall, I liked reading this work, which provides a clean treatment of the subject. I felt that the topic and results were somehow on the simple side though, both in themselves and in its similarity to the previously treated classification case. However, I do not have strong objections against acceptance, should a consensus emerge for it. ---Update--- The authors answered my questions well, in particular on the need for convergence rates in L-infty norm for k-NN (extending L-infty consistency and L-2 convergence rates found in standard references). Although the complexity of the problem still feels a bit limited, I raised my score accordingly.


Review 2

Summary and Contributions: This paper formalizes an optimal statistical rule for rejection in regression problems, which relies on thresholding the conditional variance function. They discuss the risk and rejection rate of the proposed rule, and establish rate of convergence for the kNN predictor.

Strengths: The paper is well written. The method is described clearly, and the theoretical claims are interesting and convincing. The experiments further validate the theoretical results.

Weaknesses: 1. More motivation might be needed to understand why we need regression with rejection. 2. For the semi-supervised estimation procedure, I am wondering what is the sample size requirement for ensuring a good performance. 3. Is it possible to establish rate of convergence for a class of predictors, not just limited to kNN? 4. It is probably better to add a conclusion section.

Correctness: I did not check the mathematical proof.

Clarity: Yes.

Relation to Prior Work: Yes.

Reproducibility: Yes

Additional Feedback:


Review 3

Summary and Contributions: The paper is devoted to regression with reject option, when the abstention rate is controlled. In the framework considered, the optimal rule is exhibited, obtained by thresholding the conditional variance (see Proposition 2.1). Based on this, a plug-in approach is proposed in section 3. Beyond general consistency results, rate bounds for k-NN variants with reject options are established in section 4. The theoretical analysis carried out is illustrated by numerical experiments in section 5. All technical proofs are deferred to the Supplementary Material.

Strengths: The results of the article are well motivated, presented in a rigorous framework and compared to the state of the art.

Weaknesses: Rather than focussing on the k-NN methodology, tt would have been interesting to explain which methods can be possibly problematic when designing a variant with abstention option. This seriously reduces the interest of the contribution.

Correctness: The analysis seems correct to me.

Clarity: The paper is written with clarity and is well organized.

Relation to Prior Work: It would have been interesting to discuss how recent techniques for regression error estimation, such as those analysed in Devroye et al. (2018), could be used for dealing with regression with abstention.

Reproducibility: Yes

Additional Feedback:


Review 4

Summary and Contributions: I read the feedback and do not satisfy it in answering my questions, such as question 5 below, such as the one below. The authors did not answer it for unknown reason. In fact, this point is critical for showing the effectiveness of the proposed approach. This papers presented a study of logistic regression with a reject option. They considered an optimal predictor in terms of minimum error with a constraint of fixed rejection rate. They derived the optimal rule which relies on thresholding the conditional variance function. They proposed plug-in estimations for the cumulative distribution, and reject ratio-predictor. For the specific case of kNN based estimations, they provided the rates of convergence in relation to the reject ratio. In the numerical study, they adopted two benchmark datasets to demonstrate the performance of their approach, with different reject ratios.

Strengths: I consider the authors proposed a plug-in approach for binary classification with bounded abstention. The authors used two data sets for the estimations, which seem to be reasonable and applicable in applications. They derived KNN based predictor with reject option and its rate convergence using the three assumptions. The result seems to be novel. The numerical study showed the rejection rates were close to their expected values by the proposed approaches.

Weaknesses: This paper was missing a comparative study with the existing works. I list three papers below which may be useful for reading (I know those authors only by reading their works). On the use of ROC analysis for the optimization of abstaining classifiers T Pietraszek - Machine Learning, 2007 - Springer Classification with a reject option using a hinge loss PL Bartlett, MH Wegkamp - Journal of Machine Learning Research, 2008 - jmlr.org Binary Classification with Bounded Abstention Rate S Shekhar, M Ghavamzadeh, T Javidi - arXiv preprint arXiv:1905.09561, 2019 - arxiv.org The last paper is close to this study but appeared on arXiv. I list it only for the authors' awareness.

Correctness: Most parts are reasonable for me. For the points in the Broader impact, I consider a reject option is a human strategy for the reason of reducing errors in decision. Thus the error rate is mostly concerned than the reject rate in the reject option. The approach using a reject rate, instead of error rate directly, as a bound seems a kind of weakness. I am a bit of confusion about "the main drawback of our approach is that border instances may be automatically treated while they would have deserved a human consideration".

Clarity: Yes.

Relation to Prior Work: This work is insufficient to cite the prior works, and does not discuss the differences from the existing works. For example, the author gave an equation of the excess risk. However, a similar equation called the regret was appeared in [10]. What is the difference? I prefer to see some numerical experiments in comparison with the existing works if possible.

Reproducibility: No

Additional Feedback: 1. I don't know why authors do not use the conventional way to the problem. They present a new framework on Page 2, but I get difficulty to follow it. Say, the second equation on Page 2, is it the same as that, Ld(f), in Reference [10]? If so, it is better to cite it. If not, it is necessary to tell the difference. For example, in [10], they show the relations 0 ≤ d ≤ 1/2. In this paper, the authors give the constraint lambda > 0 without an upper bound. They stated that ``lambda can be interpreted as the price to pay to use the reject option." But why it does not have an upper bound? If the price to pay for an error is 1 in an implicate sense, the price of reject can go higher than 1? The figure 1 in this paper is similar to the figure 1 in [10]. I can understand the d parameter in [10], but not for lambda (from 0 to 60) in this paper. The figure 1 in this paper does not use error rate, and it is strange to see the error curve is not shown with the same scale with the reject curve. 2. If some equation is from other work, the authors are better to cite it. The authors developed the plug-in rule which was also appeared in the papers, say, Bartlett and Wegkamp (2008). A comparison or discussion should be given about the differences between the proposed one and the existing study. 3. Theorem 3.2. I do not understand the statement for "Interestingly, this result applies to any consistent estimators of ...". Does this means the same order result for the two estimators? 4. Page 7, Line 282. It seems to be 1-epsilon rather than epsilon. 5. Numerical solutions were not so convincing when 1-epsilon -> 1. Say, Figure 3, the plot of KNN on the airfoil dataset did not show the tendency to zero error. Figure 4 showed the similar situation.

[Author Response · NeurIPS 2020]

We thank all reviewers for their valuable comments. We will address all the minor points (typos, notation issues, and remarks) as underlined and requested by the referees. Hereafter, we address comments shared by several reviewers.

**Generalization beyond $k$NN method.** First of all, although we apply our methodology to the $k$NN algorithm, we provide general consistency results in Theorem 3.2 for the proposed plug-in procedure that apply to *any* off-the-shelf estimators on the regression and the variance functions provided that these estimates are consistent. We illustrate this capacity in particular in the numerical experiments on popular machine learning algorithms such as random forests or svm for instance.

**Additional technical aspect.** As just mentioned, our methodology can be used to any estimators of the regression and the variance functions. However, our general Theorem 3.2 asks for the continuity of the cumulative distribution function of $\hat{\sigma}(X)$ (see Assumption 3.1). This assumption is violated when $k$NN algorithm is used and this is why, in addition to its popularity, we specify our methodology in the context of $k$NN. In particular Section 4.2 presents a randomized predictor based on $k$NN algorithm. This randomization technique used to circumvent Assumption 3.1 is rather simple but we believe that it is instructive from a methodological aspect. Even if the considered estimators of the variance function satisfies Assumption 3.1 the randomization technique can also be applied. In some sense, the construction given in Section 4.2 makes our results more general. Besides, we mention that Assumption 3.1 is one of the limitation that appears in Denis and Hebiri (2019) in the classification with reject option framework, and then Section 4.2 provides a way to handle this issue.

**Extension to the regression setting with application to $k$NN.** We totally agree with **Reviewer 1**: most of our results have direct analogues for classification (we will add more precise pointers to the final version upon acceptance). Considering the regression setting is an interesting application of the reject option problem which is new in the literature up to our knowledge and actually helps to understand a bit more the different characteristics of the model that are relevant for the rejection rule such as the variance function (which is intuitive in the end but good to be noticed). Moreover Section 4 is an original application to $k$NN that has no analogue and poses some additional technical tools since *i)* Assumption 3.1 does not hold with $k$NN and then randomization is required; *ii)* it asks for a finite sample bound on the $L_\infty$-norm estimation error of $k$NN which we derive based on previous asymptotic works on $k$NN.

$L_\infty$**-norm bound.** The obtained rates of convergence relies on Proposition C.1 (given in the supplementary material), it requires rate of convergence for the estimation of the variance function with respect to the $L_\infty$-norm. In particular, we establish that

$$\mathbb{E}\left[\left(\sup_{x \in \mathcal{C}} \left|\hat{\sigma}^2(x) - \sigma^2(x)\right|\right)\right] \leq C \log(n) n^{-1/(d+2)} \ .$$

This result relies on the rate of convergence of $\hat{f}$ (the estimator of the regression function) *w.r.t.* $L_\infty$ norm and can be easily extended for instance to the class of partitioning estimators or kernel estimators, under similar assumptions as in Section 4. That is to say the rate of convergence given in Theorem 4.4 applies to estimators which can be written as $\hat{f}(x) = \sum_{i=1}^{n} W_{n,i}(x) Y_i$ where $W_{n,i}$ are weight functions. We also want to point out that we do not find a generic reference for the rate of convergence for $k$NN estimates in $L_\infty$-norm. But, the proof of this result shares ideas similar to the proof of Theorem 12.1 in Biau and Devroye (2015) which establishes only the consistency of the $k$NN estimates *w.r.t.* $L_\infty$-norm.

**About the sample size of the semi-supervised procedure.** First, we want to notice that the result provided in Theorem 4.4 shows, from a theoretical perspective, the dependency with respect to the sample size of labeled and unlabeled sample. From the practical point of view, moderate values of $N$ ($N \propto 100$) already describe a regime of convergence for the estimation of $F_{\hat{\sigma}}$. We are convinced that numerical performance of the proposed semi-supervised method is mainly dictated by the quality of the estimation of the regression function and the variance function which is one of the main issues of the regression with reject option problem.

**Bibliographic remark.** Finally, we will include the references to the classification with reject option shared by the referees. In particular, we will add in the text the reference to the paper of Herbei and Wegkamp (2006) to underline the relation with the classification with reject option framework. We also want to highlight that, since we focus on the regression setting, the variance function $\sigma^2$ is not necessarily bounded and thus we do not specify any upper bound for $\lambda$ in the paper.

[Meta-Review · NeurIPS 2020]

There is a positive consensus on the contributions of the submission, with some reservations, including (R1,R4) that the theoretical results do not appear to be technically extremely novel compared to the classification case. R4 also formulates some other formal criticisms. I (weakly) recommend for acceptance of the paper.